# Somatic whole genome dynamics of precancer in Barrett's esophagus reveals features associated with disease progression

Thomas G. Paulson [1,9✉], Patricia C. Galipeau [1,9], Kenji M. Oman[1], Carissa A. Sanchez [1], Mary K. Kuhner[2,3], Lucian P. Smith [2], Kevin Hadi[4], Minita Shah[4], Kanika Arora[4], Jennifer Shelton[4], Molly Johnson[4], Andre Corvelo[4], Carlo C. Maley [5], Xiaotong Yao[4], Rashesh Sanghvi [4], Elisa Venturini[4], Anne-Katrin Emde[6], Benjamin Hubert[4], Marcin Imielinski [4,7], Nicolas Robine [4], Brian J. Reid[1,2,3,8] & Xiaohong Li[1✉]

While the genomes of normal tissues undergo dynamic changes over time, little is understood about the temporal-spatial dynamics of genomes in premalignant tissues that progress to cancer compared to those that remain cancer-free. Here we use whole genome sequencing to contrast genomic alterations in 427 longitudinal samples from 40 patients with stable Barrett's esophagus compared to 40 Barrett's patients who progressed to esophageal adenocarcinoma (ESAD). We show the same somatic mutational processes are active in Barrett's tissue regardless of outcome, with high levels of mutation, ESAD gene and focal chromosomal alterations, and similar mutational signatures. The critical distinction between stable Barrett's versus those who progress to cancer is acquisition and expansion of TP53−/− cell populations having complex structural variants and high-level amplifications, which are detectable up to six years prior to a cancer diagnosis. These findings reveal the timing of common somatic genome dynamics in stable Barrett's esophagus and define key genomic features specific to progression to esophageal adenocarcinoma, both of which are critical for cancer prevention and early detection strategies.

[1] Division of Human Biology, Fred Hutchinson Cancer Research Center, Seattle, WA 98109-1024, USA. [2] Department of Genome Sciences, University of Washington, Seattle, WA 98195-5065, USA. [3] Brotman Baty Institute for Precision Medicine, Seattle, WA 98195-5065, USA. [4] New York Genome Center (NYGC), New York, NY 10013, USA. [5] Arizona Cancer Evolution Center, Biodesign Institute and School of Life Sciences, Arizona State University, Tempe, AZ 85281, USA. [6] Variant Bio, Brooklyn, NY 11205, USA. [7] Department of Pathology and Laboratory Medicine, Englander Institute for Precision Medicine, Institute for Computational Biomedicine and Meyer Cancer Center, Weill Cornell Medical College, New York, NY 10065, USA. [8] Department of Medicine, University of Washington, Seattle, WA 98195, USA. [9] These authors contributed equally: Thomas G. Paulson, Patricia C. Galipeau. ✉email: tpaulson@fredhutch.org; xili@fredhutch.org

Normal tissues have recently been shown to harbor surprisingly extensive somatic mutations, the vast majority of which have little clinical consequence[1–6]. Barrett's esophagus (BE), a predominantly benign metaplasia that arises in the esophagus in response to chronic gastric reflux[7], also develops somatic mutations, but can further evolve extensive genomic alterations which confer significantly increased risk of progression to esophageal adenocarcinoma (ESAD)[8–11]. While important advances have been made in understanding the genomics of BE and ESAD, a key remaining question is defining molecular features, including somatic genomic dynamics, that can be used to stratify patients with BE at the highest risk of a cancer outcome (CO) vs. those likely to remain cancer free (noncancer outcome, NCO), and target those requiring aggressive treatment (ablation, endoscopic resection, surgery) vs. conservative monitoring for early detection of cancer[7].

Cancer-only studies have uncovered a vast array of genomic alterations in cancer, but are unable to provide a direct comparison of somatic genome evolution of benign neoplastic tissue in non-progressing patients from those who were ultimately diagnosed with cancer. BE is an excellent in vivo model in which to study these genome dynamics. While both BE and ESAD have very high point mutation loads, very few genes are commonly mutated across patients[10,12–14], and a large number of low-frequency gene alterations affect critical biological processes in ESAD[15–17]. ESAD is characterized by frequent somatic TP53 mutation and chromosomal copy number alterations (genome doubling [GD], aneuploidy, chromosomal instability)[9,12,13,18–20]. Complex structural chromosomal features are frequently detected in ESAD[13,21–23] and some of these events can be detected years before ESAD diagnosis[9,11,14,19,23,24]. However, the targeted, exome, and low-pass whole-genome sequencing approaches applied to date have been unable to resolve these genome-wide mutational processes and complex structural variant features in sufficient detail. To address this gap, we conducted a large-scale deep whole-genome sequencing (WGS) study of BE with a validated cancer outcome based on a longitudinal cohort. This is a unique case-control WGS study of multi-region, well-annotated longitudinal, purified endoscopic biopsies from patients with BE who have been followed without endoscopic therapeutic interventions (e.g. ablation, mucosal resection). Our WGS data, spanning 427 samples across 80 patients, allowed us to compare 40 controls with stable BE who never progressed to ESAD with 40 cases who progressed to an early, endoscopically detected incident cancer.

Here we show genomic states characteristic of BE and identify chromosomal structural dynamics common to all BE genomes. The study design allows comprehensive assessment of ESAD genes of interest in NCO and CO patients, revealing TP53 dynamics and genomic features specific to cancer risk. These results support emerging evidence that many somatic alterations detected in cancer are also detected in benign tissues and thus are not obligate for cancer progression. This study provides a valuable genomic resource and serves as a template for future pre-cancer atlas efforts[25].

## Results
**Longitudinal multi-sample study of cases with BE who progressed to an ESAD outcome compared to controls with BE who did not progress**. We sequenced whole genomes of 340 purified biopsies at high-depth (median 76X [range 40X - 106X]) as well as 62 blood and 25 normal gastric control samples at medium-depth (median 39X [range 29X - 64X]) across 80 individuals with BE (Supplementary Data File 1). These patients included 40 BE cases who progressed to ESAD ("cancer

outcome", CO) and 40 who did not progress to ESAD ("non-cancer outcome", NCO) during a median 17.47 years (range 4.46–29.63 years) follow-up period (Fig. 1, Supplementary Data Fig. 1). For each patient, we assessed two spatially mapped samples from each of two timepoints "T1" and "T2", (mean time between T1-T2 was 2.9 years in CO and 3.4 years in NCO), where T2 in the CO patients was the endoscopy in which cancer was first diagnosed (Supplementary Data File 2). We matched NCO to CO using baseline total somatic chromosomal alterations (SCA - copy gains, losses and copy neutral loss of heterozygosity (cnLOH))[19], age at T1 (T1 = first endoscopy with sufficient sample availability), and time between T1 and T2 (T2 in NCO = follow-up endoscopy randomly selected such that the distribution time between T1-T2 was similar in CO and NCO populations). In 10 NCO, we sequenced a third-time point T3, sampled a mean of 13.2 years after T1. We purified each biopsy to separate BE epithelium from the stroma and extracted DNA from purified epithelium for WGS and 2.5 M Illumina SNP array analyses (Supplementary Data File 3).

**Highly mutated clones arise and expand prior to clinical detection of BE**. Overall, the genome-wide somatic SNV + indel mutation loads per biopsy were strikingly high for precancerous tissues in both NCO and CO, with a median of 3.56 and 5.21 somatic SNV + indel mutations per megabase, respectively (Fig. 2a, Supplementary Data File 4, Supplementary Data File 38), which approaches the 6.4 SNV + indel/megabase observed in ESAD[13]. Each additional biopsy from the same patient added an average of 5446 and 8,914 unique mutations per patient in NCO and CO, respectively, consistent with extensive spatial heterogeneity of mutations reported in BE[14,26]. Per biopsy Shannon diversity, which takes into account VAF and number of genome-wide SNV and indel mutations to measure somatic genome diversity, was significantly higher in CO ($P = 0.0032$). While the overall SNV + indel mutation burden per biopsy ($P = 0.004$) and per patient ($P = 0.01$) was higher in CO, neither mutation diversity nor mutation load could unambiguously separate CO from NCO (Fig. 2a, b). The higher mutation load in CO was mainly attributable to a higher load of shared mutations between biopsies, with significantly higher functional (high and moderate impact) and non-functional (low and modifier impact) shared mutations in CO compared to NCO ($P = 0.004$, $P = 0.035$, respectively) (Fig. 2c). On average, 10,157 mutations (range 20–114,484) genome-wide were shared between any two or more biopsies per patient, despite being physically separated by up to 9 cm (mean 2.27 cm).

High "trunk" mutation load (those shared by all four biopsies per patient) was observed in a majority of both NCO (24 patients, mean 4439 mutations in trunk [range 938–3,762]) and CO (26 patients, mean 7,741 in trunk [range 834–32,558]), indicative of the early expansion of a highly mutant clone before T1 and a single clonal origin of the BE segment in most patients. In contrast, the remaining 30 patients had low (<289) trunk mutations, consistent with a multiclonal origin or early divergence of clones during the establishment of the BE segment. Trunk mutations were detected in at least one patient in 1254 genes across these 24 NCO and 26 CO patients (Supplementary Data File 7), with functional trunk mutations significantly less frequent in NCO (mean 15.5 genes/patient [range 1-61]) compared to CO (mean 39.7 genes/patient [range 3–177]) ($P = 0.040$). Functional trunk mutations included 39 ESAD associated genes (in 12 NCO and 21 CO)[10,13,27] (Supplementary Data File 8) and 17 out of 66 gastroesophageal driver genes (in 7 NCO and 17 CO) identified in Dietlein et al.,[28]. This indicates highly selected mutations can arise long before the onset of

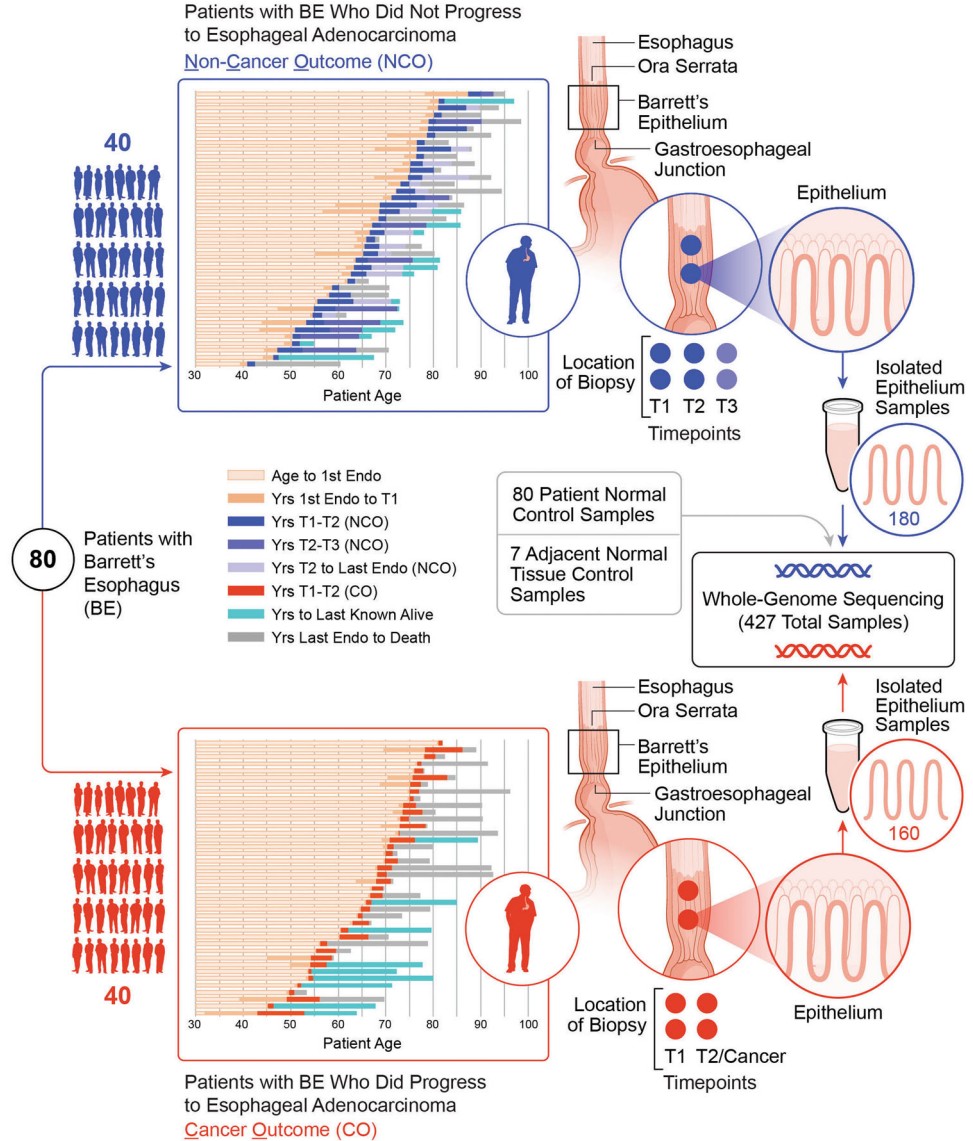

**Fig. 1 Longitudinal multi-sample study in cases with BE who progressed to an ESAD outcome compared to controls with BE who did not progress.** Schematic of our study including 340 spatially mapped BE biopsies and 87 normal control samples across 80 patients with diagnosed BE, including 40 controls with BE who did not progress to ESAD and 40 cases who progressed to an endoscopically detected, incident ESAD. Unless otherwise noted, results combine T1 and T2 time points and do not include NCO T3 or the seven additional adjacent normal gastric control samples. Source data are indicated in Figure Source Data File.

ESAD, as well as early in BE tissue that does not progress to ESAD over long follow-up.

Despite evidence for the spatial spread of clones, mutations private to single biopsies were more numerous than shared mutations in both NCO and CO (Fig. 2d), indicating ongoing mutational processes during evolution. The pairwise divergence between biopsies was highly variable across patients and did not significantly distinguish NCO and CO, regardless of mutation classes (i.e., functional/nonfunctional, clonal/subclonal). We directly compared total mutation load with both patient age at the time of biopsy and time between T1 to T2 and found no significant differences in either NCO or CO. Using an EM algorithm to infer the average change in mutation load between T1 and T2 (see Methods), we found no significant change in NCO ($P = 0.9$), but a small, significant average increase of 947 mutations/year per biopsy (approximately 6.5% of the median mutation load of a CO biopsy) accumulated between T1 and T2 in CO ($P = 0.012$, 95% CI 128–1,766). Taken together, these results

suggest that in most patients with BE, independent of ESAD progression, there is early clonal expansion of a highly mutant progenitor, including mutations in ESAD-associated genes, with continued localized mutation accumulation.

Mutation signatures reflect the combination of intrinsic mutagenic processes and extrinsic exposure to mutagens over the history of a tissue[29,30]. To determine whether BE biopsies from CO show evidence of distinct mutagenic processes relative to NCO, we used SigProfiler to extract single-base (SB) signatures across all biopsies. We detected 10 Cosmic SB signatures, nine of which were previously identified at similar proportions in ESAD[29] (Fig. 2e, Supplementary Data File 9). Overall, we found no significant difference in detection of each signature by the patient between CO and NCO for each of the ten signatures (adjusted p values all >0.05); by biopsy, we found SBS18 was detected marginally more frequently in NCO ($P = 0.041$) and SBS40 marginally more frequent in CO ($P = 0.041$). SBS17a/b (unknown etiology; common in the

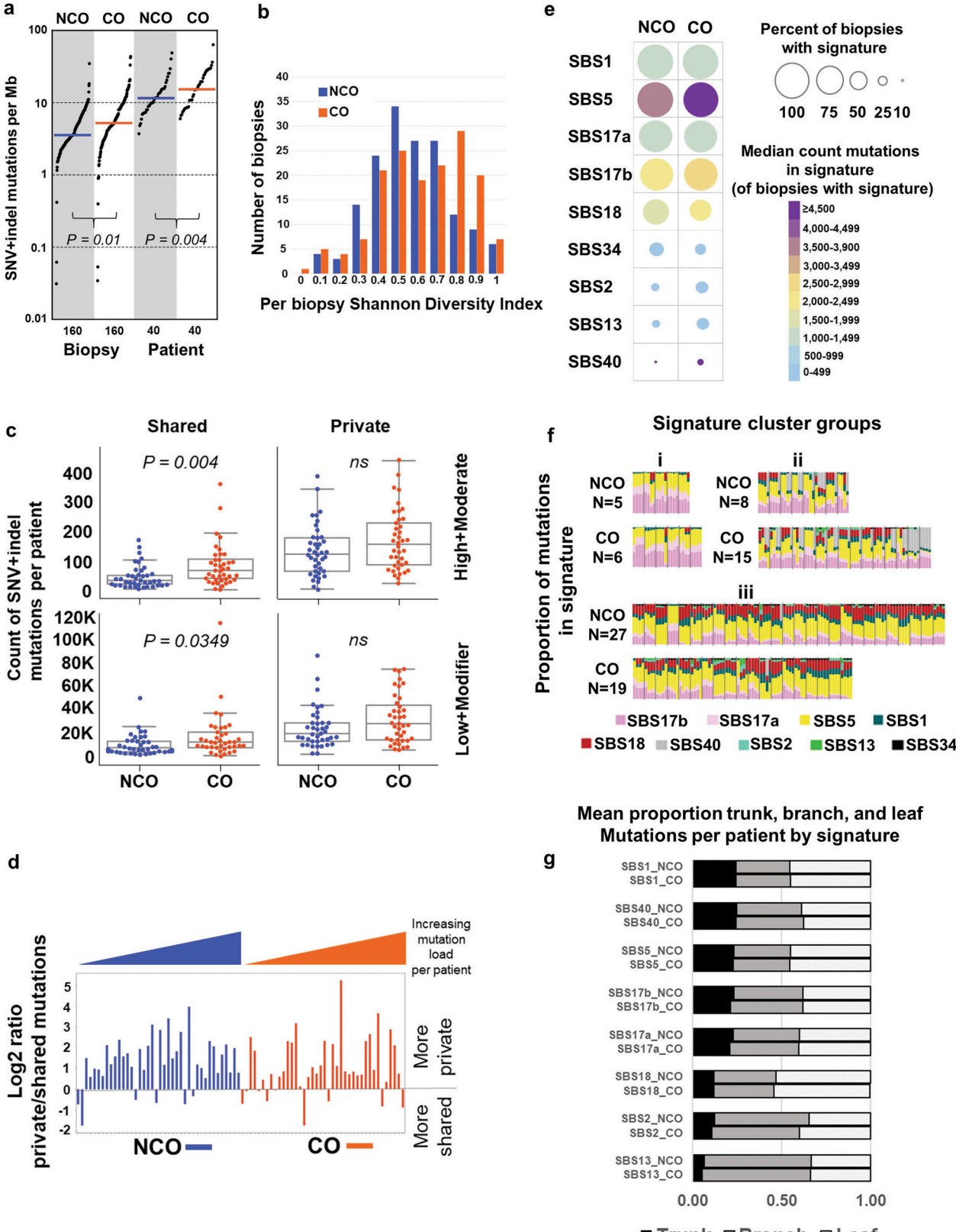

stomach and esophageal adenocarcinoma) and SBS5 (unknown etiology; correlated with age) were nearly ubiquitous across patients and samples, and typically had high numbers of assigned mutations in all four biopsies per patient. Previous studies have consistently detected SBS17a/b in both BE and ESAD[10,12,14,27,31–33]. Our results further demonstrate this

mutation signature was not associated with cancer outcome, but rather with the tissue environment in which BE develops. The proportion of both SBS17a and SBS17b increased significantly with increasing single base mutation load in both NCO and CO (P < 0.0001), and has also been previously observed in ESAD[10].

**Fig. 2 Highly mutated clones arise and expand prior to clinical detection of BE. a** y-axis (log scale) shows unique SNV + indel mutations per megabase (2,800 Mb of sequence) per biopsy and per patient in NCO and CO. Per patient mutation burden was derived from the sum of unique SNV and indel mutations across four biopsies. Horizontal bar is median (3.56 vs. 5.21 by biopsy, and 11.62 vs. 15.35 by patient, in NCO and CO, respectively). Nine biopsies with exceptionally low mutation load were also included (Supplementary Data File 5, See "Anomalous biopsies" in Supplementary Methods). **b** Mutational diversity per biopsy using Shannon Diversity Index based on SNV + indel load and VAF in NCO patients (blue) and CO patients (orange). **c** Count of unique SNV and indel mutations per patient classified as shared between biopsies (left) or private to a single biopsy (right), with "functional" (high or moderate) impact on protein function based on snpEFF[92] (top) vs. low or modifier (bottom) (comparison by t-test), (Supplementary Data File 6). Box centerline indicates median, box edges 1st and 3rd quartiles and whiskers 1.5x interquartile range (IQR), all data points outside of IQR were plotted. A total of 40 CO (160 biopsies) and 40 NCO (160 biopsies) patients were examined. Mann–Whitney U test was used for comparison of the two groups. **d** Log2 ratio of private/shared SNV and indel mutations by patient. Patients with more mutations private to single biopsies than shared between two or more biopsies have values above zero; those with more shared mutations than private have values below zero. **e** Percent of biopsies with mutation signatures (circle size) and median number of mutations per biopsy in each signature, including only biopsies in which that signature was detected (color scale). SBS3 not shown(only detected in a single CO biopsy). **f**, Signature cluster groups where each column is a single biopsy. Biopsies are grouped by patient and ordered by patient ID. N = count of NCO or CO patients in each signature cluster. Comparison by Fisher's exact test. **g** Signature probabilities were assigned to each mutation, mutations were binned as trunk (shared by all four biopsies per patient), branch (shared by 2 or 3), or leaf (private to a single biopsy), and the mean proportion of mutations in trunk, branch, or leaf for each signature per patient was calculated. SBS3 and SBS34 not shown (very low mutation counts), see Supplementary Data File 10. Source data are indicated in Figure Source Data File.

We examined whether combinations of mutation signatures separated patients based upon their progression status. Hierarchical clustering using cosine similarity of mutation signatures resulted in three clusters of patients (Fig. 2f): i) ubiquitous and a high count of SBS5 and SBS17a/b, ii) ubiquitous SBS1, SBS5, SBS17a/b, plus one or more biopsies with SBS40, and iii) ubiquitous signatures plus a combination of SBS18, SBS2/SBS13, and SBS34. However, the count of NCO vs. CO patients in each cluster was not significantly different ($P = 0.999, 0.076, 0.144$ for clusters 1, 2, and 3, respectively). In both NCO and CO, roughly half of the patients had variable mutation signatures between biopsies, consistent with ongoing clone-specific or localized mutagenic insults within individual patients (Supplementary Data Fig. 2). Of the eight signatures with sufficient numbers of mutations for evaluation, SBS1, SBS40, SBS5, and SBS17a/b had similar mean proportion of mutations per patient in trunk (0.21–0.24), compared to lower mean proportion in SBS18, SBS2 and SBS13 (0.05 - 0.12) (Fig. 2g, Supplementary Data File 10), suggesting SBS18, SBS2 and SBS13 arise more often as localized or "later" events during the evolution of the BE tissue. Overall, mutation signatures are very similar between NCO and CO, suggesting the exposures that cause them are characteristic of BE rather than specific to ESAD development.

Chromosomal alterations at fragile sites[19,27,34] were ubiquitous, with all 80 patients having at least one sample with somatic structural variants (SVs) affecting *WWOX* and *FHIT* (Supplementary Data File 11). Rigma[23] (densely clustered, low copy-number deletions) was the most common complex SV type affecting these genes, observed at *WWOX* in 43% and *FHIT* in 87% of biopsies (Supplementary Data Fig. 3). Overall, 88.6% of rigma events occurred in known or suspected fragile sites. Clonal rigma events (occurring in >1 sample per patient) were found in 30 NCO patients and 34 CO, indicating these events typically arose before clinical detection of BE. The third most common gene affected by SVs, *TTC28*, was altered in nearly all NCO and CO patients (39/40 and 37/40, respectively) (Supplementary Data Fig. 3). *TTC28* contains a promiscuous LINE-1 retrotransposon, previously characterized in colorectal, esophageal, and other cancers[35,36]. CO had significantly more biopsies with multiple events throughout the genome involving *TTC28* than NCO ($P = 0.027$), perhaps reflecting increased tolerance for genomic instability in patients who progress to ESAD.

**Somatic alterations in ESAD genes and mutation selection before cancer**. Multiple studies have highlighted genes of interest that are frequently altered in ESAD[10,13,27]. To assess differences

in these genes between NCO and CO, we compiled a list of 127 ESAD associated genes from these studies and quantified functionally significant high and moderate impact point mutations and indels, homozygous deletions (HD), high-level copy gain, and structural variations (SVs) (Supplementary Data File 12). Fifteen of these genes (11.8%) were altered significantly more frequently in CO compared to NCO (FDR < = 0.1), including *TP53*, *APC*, *CDK12*, *ERBB2*, *PCDH17*, and *GATA6*, and three were altered at higher frequency in NCO (*C6orf118*, *CDKN2A*, and *AGBL4*) (Fig. 3a, Supplementary Data File 8). While some of these frequently altered ESAD genes may be inherently prone to excessive mutations and thus have low evidence for selection[28,37], our study design allowed a direct comparison of CO and NCO populations to distinguish between inherent instability vs. being selected for the development of cancer. When only considering SNV + indels, patients with CO had a significantly higher cumulative burden of these mutated ESAD genes per patient (median of 9 vs. 5 ESAD genes mutated, $P = 0.0004$); however, the vast majority (109 of 127, 85.8%) were mutated at similar frequencies in NCO compared to CO, including frequently altered ESAD genes such as *ARID1A*, *SMARCA4*, and *SYNE1*. Although statistically different, the overlap of distribution in the burden of mutated ESAD genes renders this a poor discriminator between NCO and CO patients (Fig. 3b). Mutations in ESAD genes were more likely to be shared across biopsies (i.e., an expanded clone) in CO compared to NCO ($P = 0.047$). Strikingly, a genome-wide examination of genes under positive selection using dN/dS ratios[37] showed nearly all genes displaying evidence of selection were selected in both NCO and CO (Fig. 3c, Supplementary Data File 13). Evidence of convergent evolution (multiple functional mutations in the same gene in different samples from a single patient) was found in 24 ESAD genes in 20 NCO and 27 CO patients, with six NCO and eight CO patients having three or more functional mutations in the same gene, suggesting selection for mutations was mostly irrespective of progression to ESAD (Fig. 3d-g, Supplementary Data File 14). These data argue against the assumption that positively selected mutations invariably indicate a cancer-promoting process and suggest that many putative cancer "driver" genes may be under selection for properties that don't necessarily lead to cancer[38]. In BE, this may manifest as selection during wound healing in response to chronic acid injury or exposure to mutagens in the bile refluxate that occur long before the development of a cancer.

**Gene alterations selected in CO patients**. Patients with CO had significantly more functional mutations in 44 genes compared to

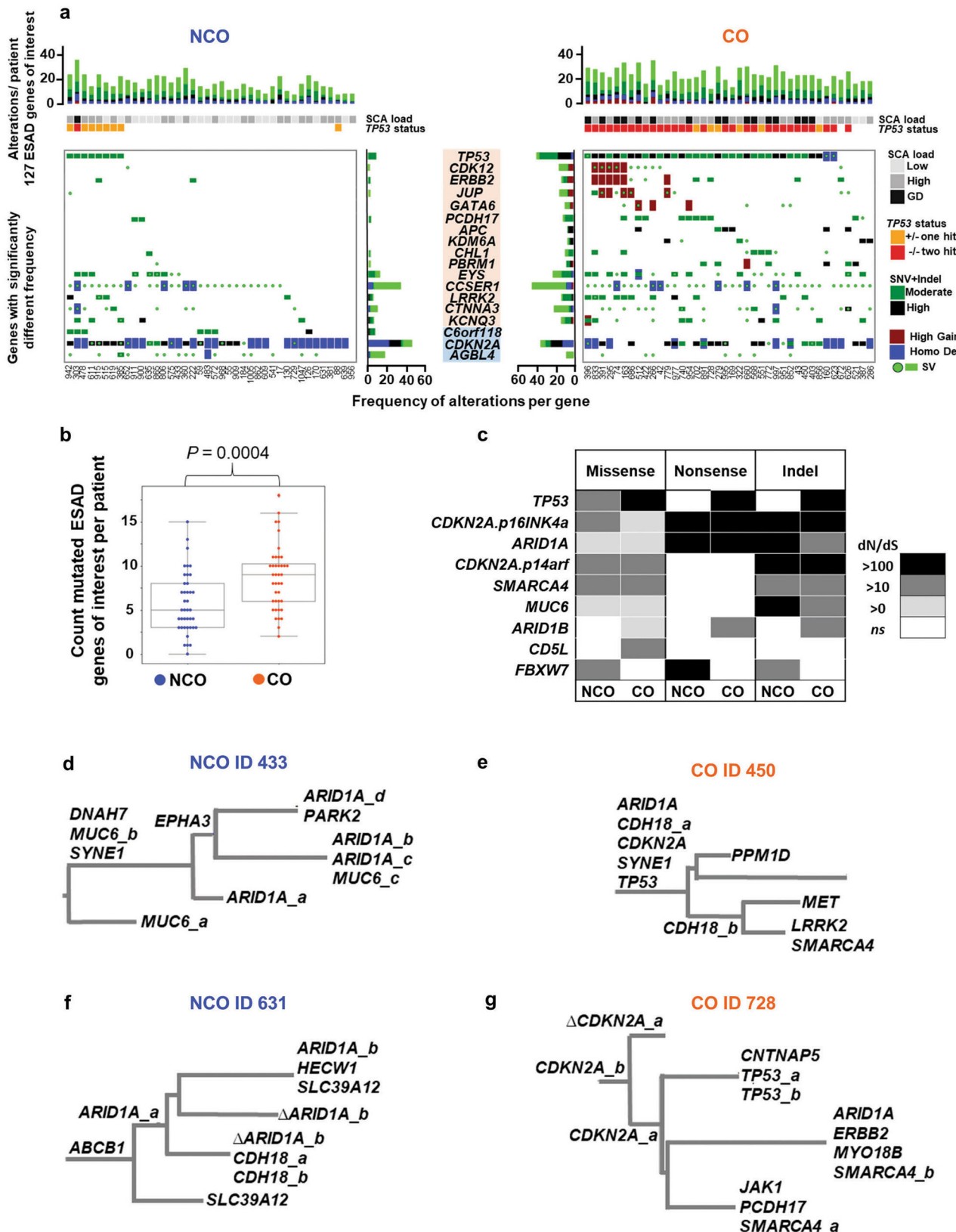

NCO (<=0.1 FDR), with the ten most significant being *TP53*, *RYR3*, *PCDH8*, *SPATA31D1*, *APC*, *CACNA1H*, *PBRM1*, *SZT2*, *BRINP3*, and *ZNF568* (Supplementary Data File 15 and see Methods); however, most of these genes were either altered in only a minority (<20%) of CO patients or were also altered in NCO. Genes important for progression to ESAD may not be revealed if causal mutations occur only at low frequency in any one gene, but rather are spread across many genes in a pathway or a process[15]. We conducted a comprehensive gene pathway analysis to evaluate the effects of the individual mutations (SNVs/indels and double deletions) on pathway function (Supplementary Data Files 16, 17, and 18); copy number and SV alterations

**Fig. 3 Somatic alterations in ESAD genes and mutation selection before cancer. a** Top panels show cumulative functional alterations in 127 ESAD genes of interest per patient, with patient ID indicated below maximum SCA load and *TP53* status per patient. Bottom panels show alteration types and frequency per gene in CO (orange) and NCO (blue) for genes with significantly different frequencies of alterations between CO and NCO (FDR < = 0.1) (Supplementary Data File 8). **b** Count of mutated ESAD genes per patient. Box centerline indicates median, box edges 1st and 3rd quartiles and whiskers 1.5x interquartile range (IQR), all data points outside of IQR were plotted. A total of 40 CO (160 biopsies) and 40 NCO (160 biopsies) patients were examined. Mann–Whitney U test was used for comparison of the two groups. **c**, Nine genes had a significant dN/dS ratio >0 in NCO and/or CO patients. dN/dS reflects the fraction of mutations observed in a gene that are likely to be under positive selection[37]. A dN/dS of 10 indicates that there are 10 times more non-synonymous mutations in the gene than neutrally expected, suggesting that at least around 90% of mutations in that gene are selected. **d**–**g** Parsimony tree phylogenies from typical BE patients having an average number of altered ESAD genes per patient, showing examples of heterogeneity of mutations in ESAD genes of interest mutations and convergent evolution in both NCO and CO. Only ESAD genes of interest are annotated, and Δ indicates the haplotype with the mutated allele identified in another sample was lost due to a copy loss event in this branch. Branch lengths are proportional to inferred mutation count for all SNVs. Mutations suffixed _a, _b, etc., indicate mutations at different sites in the same gene. Annotated phylogenies for all patients can be found in Supplementary Data File 36. **d** NCO patient with three *MUC6* mutations (i.e. *MUC6_a, b, c*) and four *ARID1A* mutations. **e** CO patient with two *CHD18* mutations. **f**, NCO patient with two different *ARID1A* mutations and two *CHD18* mutations. **g** CO patient with two *CDKN2A* mutations, two *TP53* mutations, and two *SMARCA4* mutations. Source data are indicated in Figure Source Data File.

were not included as their effects on gene function are more difficult to determine. Twenty-seven pathways were altered at significantly higher frequency in CO compared to NCO. Given that *TP53* alterations are highly enriched in CO, this analysis was repeated without including *TP53* mutations to identify significantly altered pathways without considering *TP53* status. With *TP53* excluded, five pathways remained significant, with one, the "Presenilin action in Notch and WNT signaling pathway" having significantly more mutations per patient in CO and more CO patients with the pathway altered (Supplementary Data Fig. 4, Supplementary Data Files 19 and 20). Presenilin has been shown to regulate the Wnt/beta-catenin signaling pathway[39] and to have regulatory effects on beta-catenin turnover and tumorigenesis[40]. Our results suggest a potential role for alterations in genes in this pathway (e.g. *APC*, *JUN*, *CREBBP*, *MYC*, and *CTNNB1*) in progression to ESAD.

We detected structural variant (SV) events at significantly different frequencies in 108 genes between NCO and CO, with 94.4% (102/108) higher in CO (Fisher's exact test adjusted for multiple comparisons). Six genes (5.56%) clustered on chromosomes 17q spanning 37.6 - 40 Mb (*CDK12*, *IKZF3*, *THRA*, *CCR7*, *JUP*, *DNAJC7*) and 25 genes (23.15%) clustered on 18q (e.g., breakpoint within *MIB1*, or amplification of region around *GATA6*) were disrupted preferentially in CO, while five of the six loci with significantly higher SVs in NCO were concentrated around *CDKN2A* on chromosome 9p (Fig. 4, Supplementary Data File 21). These data suggest some genome regions, e.g. fragile sites, were susceptible to SV disruption in all BE patients; however, in CO the development of SVs was more frequent and extensive, suggesting additional selection for alterations in genes that provide some growth advantage[41]. Finally, high-level amplifications of ESAD genes were observed almost entirely in CO patients and included genes such as *MYC*[21], *CDK12*[42], *ERBB2*[10] and *GATA6*[20,43] (Supplementary Data File 22). Altogether, while we found many gene-specific alterations in BE patients that have been reported in ESAD specific studies[10,13,15,20,21], those altered significantly more often in CO were usually found in only a minority of patients, with the notable exception of *TP53*.

**Expansion of cell populations with two-hit *TP53* (−/−) is a primary characteristic for progression to ESAD.** *TP53* alterations are strongly linked with progression to ESAD and can be detected years before ESAD diagnosis[9,18,33,44,45], yet focal microscopic regions with *TP53* mutations have been found to be remarkably common in normal tissues[2,46,47] including in normal squamous esophagus[5,6]. We detected likely functional *TP53* mutation, homozygous deletion (HD), or single-copy loss in at

least one biopsy in 90% (36/40 patients, 98/160 biopsies) of CO, but only in 22.5% (9/40 patients, 14/160 biopsies) of NCO (Fig. 5a). The *TP53* mutation spectrum was similar between NCO and CO, with 8/9 NCO mutations being reported in the IARC *TP53* mutation database[48], within the DNA binding or oligomerization domains, or with a FATHMM score > =0.94, suggesting high pathogenicity (Supplementary Data File 23). Nevertheless, NCO patients with such lesions in *TP53* remained cancer-free throughout endoscopic surveillance for an average of 10.09 years, and for an average of 4.94 years after the mutations were detected.

Evaluating spatially separated samples collected over multiple time points revealed CO were more likely to have bi-allelic inactivation of *TP53* and to have had the mutated clone spread in the esophagus compared to NCO. The 112 biopsies with *TP53* alterations were classified into "one-hit" (*TP53* +/−, 12 NCO, 23 CO biopsies) or "two-hit" (*TP53* −/−, 2 biopsies in a single NCO, ID 303; 75 CO biopsies). *TP53* −/− was detected in 75% (30/40) of CO, compared to only 2.5% (1/40) of NCO (Fig. 5b). Among patients with *TP53* mutations, a shared (clonal) *TP53* alteration, which had spread to multiple biopsies was detected in 75% (27/36) of CO but only 33% (3/9) of NCO (P = 0.04). *TP53* mutations in CO had higher average VAF within biopsies (0.64 vs. 0.346, P < 0.0001) and were found in more biopsies per patient (2.7 vs. 1.8, P = 0.00472) compared to NCO. Significantly more CO had *TP53* mutations at both timepoints (25/36), compared to NCO (2/9, P ≪ 0.001) (Supplementary Data File 24), consistent with other studies showing *TP53* alterations are frequently early events in patients who progress to ESAD[9,45,49]. Thus, while potentially pathogenic *TP53* mutations were detected in non-progressing BE, they were generally subclonal, one-hit, and localized, compared to *TP53* mutations detected in BE patients who progressed to ESAD.

Four CO patients did not have mutations, deletions or SVs affecting *TP53* in any of the four biopsies evaluated (IDs 286, 387, 521, 672, Supplementary Data Fig. 5). One common feature among these four patients was chromosome (chr) 18q copy gain spanning 18-22 Mb, which has been shown to be associated with increased risk of progression to ESAD more than four years before ESAD diagnosis[11]. This region includes *GATA6*, a transcriptional activator important for determining gastrointestinal cell fate during embryogenesis which has been proposed to play a role in the development of BE and progression to ESAD when present at increased copy number[20,50,51]. Across all patients, biopsies with *GATA6* chromosome copy gain (>=3 copies) had significantly higher SCA load (copy number gains, losses and cnLOH[19]) (P ≪ 0.001), as well as significantly higher SV load when occurring with *TP53* alterations (P ≪ 0.001),

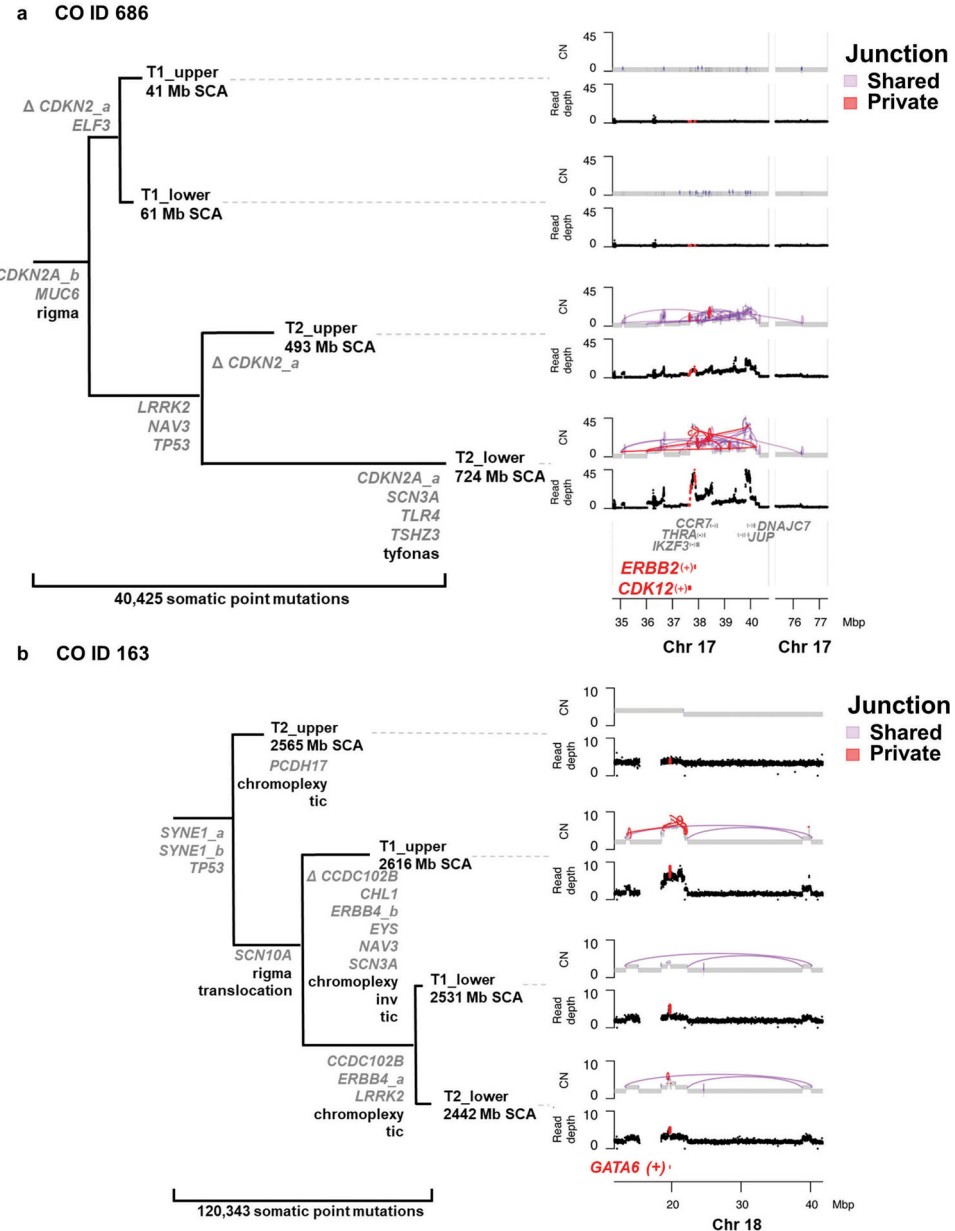

**Fig. 4 Gene alterations within complex SV events selected in CO patients. a** CO ID 686 with complex SV in both T2 samples, which cluster together in the SNV-based parsimony tree. The phylogeny is annotated with ESAD genes of interest with functional 2+ caller mutations (multiple mutations in the same gene are appended with _a, _b), and Δ indicates chromosome copy loss of mutated allele. The complex structural rearrangement pattern of rigma[23] was detected in all four samples, whereas tyfonas ("typhoons" of high junction copy number junctions and fold back inversions[23]) was only detected in the T2 sample, shown in the right panel. Within this region, *CDK12* was disrupted by structural alterations and *ERBB2* was within one of the highly amplified regions within this complex SV. **b** CO ID 163 with expanded *TP53* −/− and genome doubling in all four samples. The three samples on the lower clade in the SNV-based phylogeny share a common amplified region which includes increased copies of *GATA6* and surrounding genes. Total Mb SCA are indicated for each sample. CN = copy number. Source data are indicated in Figure Source Data File.

compared to biopsies with no *GATA6* copy gain (Supplementary Data Fig. 6 and 7). This was especially apparent in samples with one-hit *TP53* +/- alterations. Additionally, SCA and SV load increased with increasing number of copies of *GATA6* (test for trend $P \ll 0.001$). The type of chr18 alteration differed depending upon *TP53* status. The chr18 gains in 34 of 38 samples with no *TP53* alterations were primarily whole chromosome gain or small, localized gain around *GATA6*, whereas the gains in 35 of 42 samples with *TP53* alterations had complex structural changes involving larger regions of chr18, often including loss of sequence on distal 18q. Altogether, these data suggest increased copies of a 3-4 Mb chromosomal region containing *GATA6* promotes

genome instability and progression to ESAD through increasing SCA and SV load, particularly when it occurs together with *TP53* alterations.

**Bi-allelic alteration of *TP53* drives somatic chromosome instability in progression to ESAD.** Loss of *TP53* function leads to development of multiple manifestations of genomic instability in the context of DNA damaging insults[52]. Chromosomal bridge-fusion-break (BFB) events[23] were detected in 26 CO biopsies (13/40 patients), but in only one NCO biopsy, and exclusively in the context of altered *TP53* (27/27 biopsies with BFB had altered

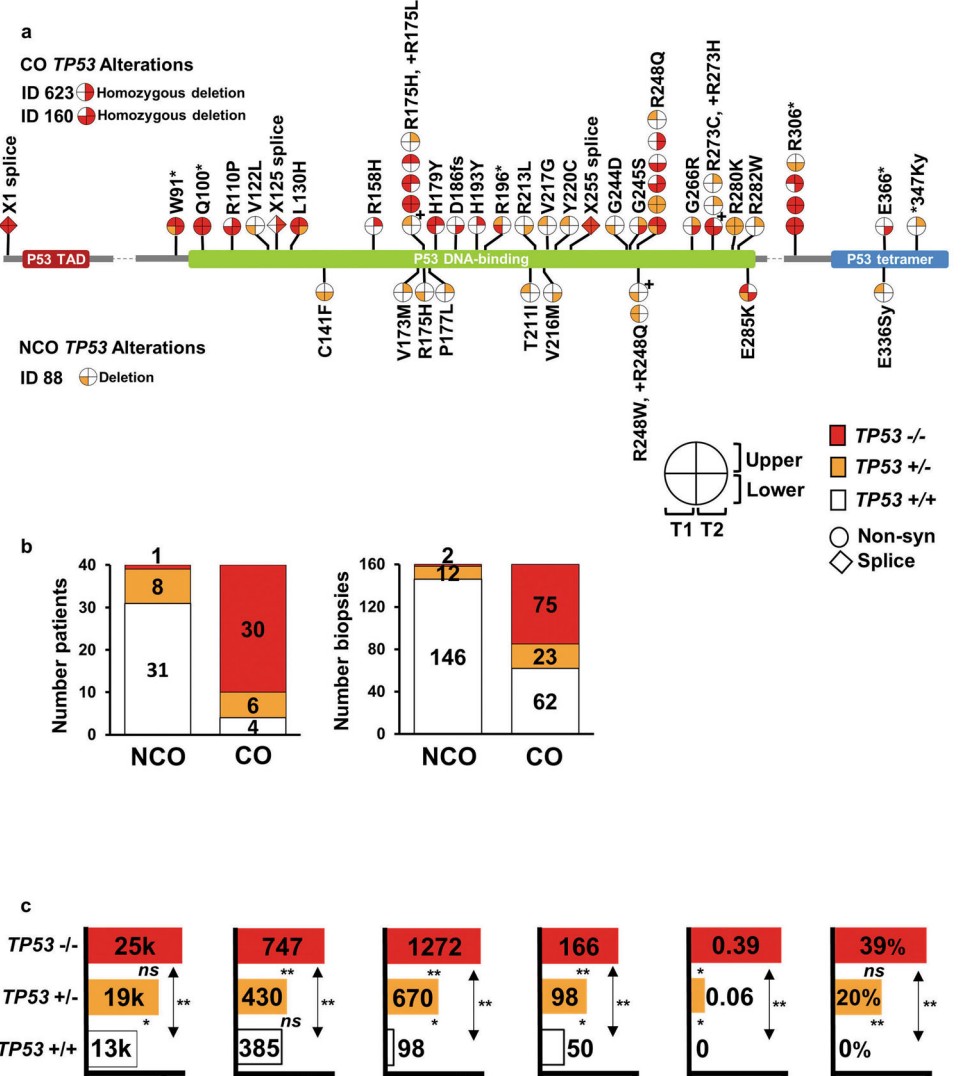

**Fig. 5 Expansion of *TP53* −/− is a primary characteristic for progression to ESAD. a** *TP53* mutations in four biopsies per patient collected over time (T1 and T2) and space (upper and lower esophagus) in CO and NCO. Each "lollipop" pie on the *TP53* gene structure indicates a specific *TP53* mutation with quadrants of the lollipop indicating the number of mutant copies in each of the patient's four biopsies. The *TP53* biopsy quadrants for two CO with homozygous deletions of *TP53* (IDs 623 and 160) and one NCO (ID 88) with single copy deletion (ID 88) spanning *TP53* are also plotted for completeness. (+) next to a pie corresponds to the amino acid change marked by (+). **b**, Proportions of patients and biopsies with wild-type *TP53* +/+ (white), one-hit *TP53* +/− (orange), and two-hit *TP53* −/− (red) status. Of the six one-hit CO, five of the mutations were private and one was shared, and of the eight one-hit NCO, six were private and two shared. **c**, Somatic alterations stratified by *TP53* zero-hit (n = 208 biopsies), one-hit (n = 35 biopsies), and two-hit (n = 77 biopsies). Two-sided Mann–Whitney U test was used for each comparison with correction for multiple comparisons. P-values between bars test for significant difference between *TP53* categories, ns = P ≥ 0.05, *=P < 0.05, **=P < 0.0001. SCA = somatic chromosomal alterations (gains, losses, cnLOH); SV = structural variants; BFB = bridge-fusion-break events. P-values and source data are indicated in Figure Source Data File.

*TP53*, P < 0.0001 for correlation of *TP53* and BFB). Statistical event ordering analysis indicated BFB and genome doubling (GD) occurred after the development of *TP53* alterations (P«0.0001). We found no evidence associating normalized telomere length, as derived from WGS data (see methods), with cancer outcome status or *TP53* alterations (Supplementary Data File 25).

We found genome doubling, common in ESAD and frequently found in BE patients who progress to ESAD[18,19,24,26,49,53,54], significantly more frequently in CO biopsies (35 biopsies, 14/40 patients) than in NCO biopsies (2 biopsies, 1/40 patients, P = 0.0003). As expected, GD was strongly associated with

*TP53* alterations (P < 0.0001), and all 37 GD biopsies were either *TP53* +/− (n = 7) or *TP53* −/− (n = 30). While GD biopsies had significantly higher total mutation load (P = 0.0043) and SV counts (P < 0.0001) compared to non-GD samples, only SV load remained significantly higher when comparing GD to non-GD *TP53* altered biopsies (P = 0.003). Nineteen genes were significantly mutated in GD biopsies compared to high instability biopsies without GD (SCA between 103.9-1,500 Mb and SV > 71) (Supplementary Data File 26), the most significant being *DCDC1*, *PPP1R3A*, and *TP53*. Biopsies with mutations in three or more of these genes had mean SCA > 1500 Mb, significantly higher than those with zero to two of the genes mutated (P = 3.92 ×10⁻¹²).

We detected GD either at T2 alone (7/14 patients) or both T1 and T2 (7/14 patients), with a mean of 2.05 years prior to cancer diagnosis, consistent with previous studies showing aneuploidy arising before ESAD in a subset of BE[9,49,53].

*TP53* −/− biopsies had significantly higher mean levels of SNVs and indels, SCA, BFB, SV, and increased occurrence of GD compared to *TP53* + / + (all *P*«0.0001) (Fig. 5C), with a significant overall trend of higher levels with increasing hits in *TP53*. We observed heterogeneity of *TP53* status within patients, with all 40 NCO and 29 CO patients having at least one *TP53* + / + biopsy. Among these *TP53* + / + biopsies, we observed no significant differences in mutation load (*P* = 0.507), SCA (*P* = 0.18), or SV count (*P* = 0.692) between CO and NCO; thus most CO patients have some BE biopsies that are indistinguishable from NCO biopsies. In *TP53* + /- biopsies, the only significant difference was that CO had a significantly higher mean SV count of 120 (median = 82) compared to 57 (median = 46) for NCO (*P* = 0.041), suggesting SV load, in the context of one-hit *TP53* alterations, may provide an additional measure to differentiate CO from NCO. The heterogeneity of *TP53* alterations across biopsies within a patient and concomitant chromosome instability within the Barrett's epithelium emphasizes the importance of multi-region analysis for accurate assessment of risk of progression to ESAD.

**Development of complex chromosomal structural variants during progression to ESAD.** WGS studies have reported SVs in ESAD[10,21] and higher SV burden in ESAD than BE[27]; we have extended these results to CO and NCO populations over space in the esophagus and over time. CO had an average of 67 more SVs per biopsy than NCO (*P* = 4.5 × 10^{-8}, 95% CI 43-91, mean NCO SVs = 54.84, mean CO SVs = 121.76) with no significant effect of patient age at sampling or position in the esophagus (Supplementary Data Fig. 8a). SV load was significantly higher in CO for the majority of chromosome arms (36/46 arms with data, Supplementary Data Fig. 8b), and rearrangement classes (Supplementary Data Fig. 8c)[55] indicating the higher load of SVs in CO was genome-wide.

To characterize complex SV features in biopsies and compare them between NCO and CO, we applied the Junction Balance Analysis tool (JaBbA[23]) (Fig. 6a). CO was correlated with the presence of double minutes (DM), BFB, translocations, template insertion cycles (TIC), and also with higher burdens of BFBs, translocations, inversions, TICs, duplications, and deletions; however, none of these comparisons remained significant when *TP53* status was included in the analysis, highlighting the critical role of the loss of *TP53*-mediated break sensing in the generation of these complex SV event features (Supplementary Data File 27). BFB events in CO were the only SV feature that was significantly more frequent at the T2 cancer diagnosis time compared to T1 (*P* = 0.0062). In both NCO and CO, rigma was the only complex SV feature detected significantly more often as a shared (early) event (*P*«0.0001) (Fig. 6b, Supplementary Data File 28, Supplementary Data Fig. 9); while maximum measures of simple and complex SVs were generally higher at T2 (Supplementary Data Fig. 10), all other SV features are more likely to be either private events or have no specific temporal pattern. Timing of SV events relative to GD showed that that rigma, inversions, TIC, and chromoplexy generally occur before GD (*P* = 0.051, 0.0021, < 0.0001, 0.0505 respectively).

SV features figure prominently in ESAD cases[23]. Using the same SV calling pipeline, we compared the SV features between NCO and CO with data from 408 ESAD cases[13,23], the vast majority of which (401/408) were treatment naive. SV burden was significantly higher in ESAD compared to CO, which was

significantly higher than NCO (Fig. 6c). Similarly, CO had a significantly higher fraction of patients with complex amplifications than NCO, yet lower fraction than in ESAD cases (Fig. 6d). Dimension reduction of SV features applied to all NCO and CO samples and to ESAD cases shows the diversity of alterations at the biopsy level, with many of the samples from CO patients that lack substantial alterations clustering with NCO samples (data not shown). This same UMAP[56] dimension reduction of SV features was applied to the most rearranged NCO and CO samples per patient, and the 408 ESAD cases, resulting in four clusters (Fig. 6e, Supplementary Data File 29). These clusters differed in their SV burden, presence of complex amplifications, and simple translocations (Fig. 6f, g). NCO were significantly enriched in Cluster 1, whereas both CO and ESAD were significantly enriched in Cluster 3 (Fig. 6h). Mutations in *TP53* and amplification of *CCNE1* (cyclin E1) were the only ESAD gene alterations found at a significantly higher frequency in a given cluster (both higher in Cluster 3, *p* = 0.0014 and *P* = 0.001, respectively). When assessing the most altered sample, CO patients were generally clustered with surgically resected ESAD, even though some samples were obtained up to 9.8 years (mean 2.81) before a small, endoscopically detected incident cancer, emphasizing the importance of complex SV features for ESAD risk assessment. No significant difference in overall survival was found for ESAD patients in different clusters.

**Specific somatic genomic features classify patients by cancer outcome.** Identifying markers of risk for progression of BE to ESAD is a key component for improving patient care. Given the sample size of our study, the complexity of the somatic genome, and with a vast number of features occurring at low frequencies, we used a multi-step approach to identify robust markers associated with CO patients. First, univariate analyses of individual SNV/indel, SV and copy number alteration data identified 47 features significantly associated with CO, including *TP53* alterations (one- and two-hit), GD, chromosome 18:18-25 Mb gain, complex SV features (BFB, DM, tyfonas), ESAD associated genes with significantly higher frequency alterations in CO, genes with significantly higher functional mutations in CO, mutated genes associated with SV features and GD, and ESAD genes with high-level amplification (Supplementary Data File 30). Next, we used a LASSO approach to reduce the number of potential features out of this group. Using a random sampling training and cross-validation procedure that left out 20% of samples for testing in each iteration, we identified the 14 features that were most frequently selected in the prediction models (Supplementary Data File 30).

With these 14 features, we then used a two-variable regression model to evaluate one-hit and two-hit *TP53* status for classifying patients as NCO vs. CO. As expected, two-hit *TP53* was strongly associated with having an ESAD outcome (*P* = 1.01×10^{-5}), while one-hit *TP53* alone did not reach significance (*P* = 0.069). Using the same multivariable modeling approach, we tested two-hit *TP53* with the 14 risk features described above (individually) and found only chromosome 18:18–25 Mb gains (*P* = 0.0081) and functional mutations in *PCDH17* (*P* = 0.018) provided independent prediction power in addition to two-hit *TP53*. We also evaluated if one-hit *TP53*, interacting with the same risk features, could add ESAD risk prediction power independent of two-hit *TP53*. Only chromosome 18:18–25 mb gains interacting with one-hit *TP53* had statistically significant predictive power independent of two-hit *TP53* using the composite features obtained from the above parsimony evaluation method. While this six feature model (one- and two-hit *TP53*, chromosome 18:18-25 Mb gain, functional mutation in *PCDH17*, and interactions between

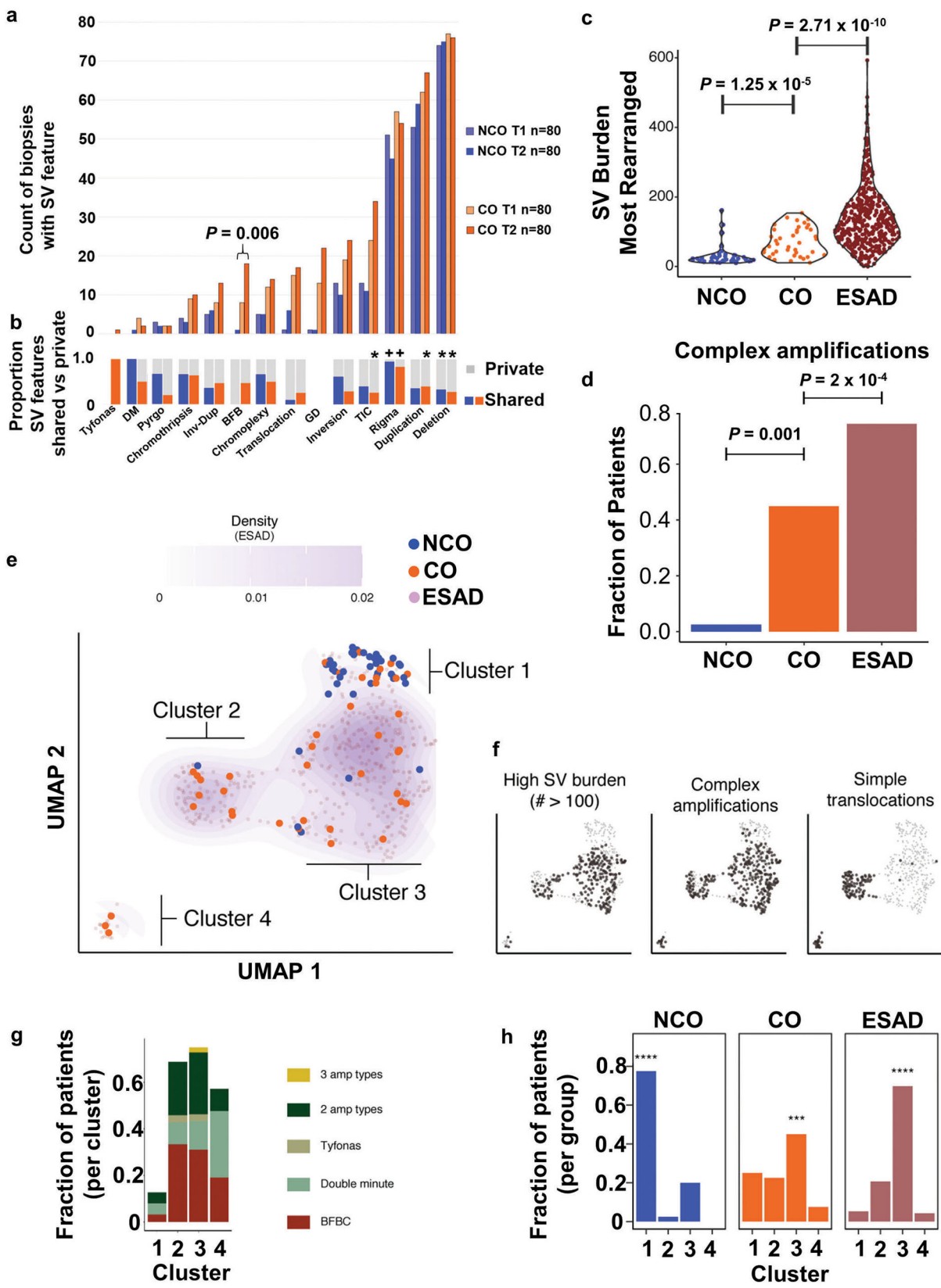

one-hit *TP53* with chromosome 18:18–25 Mb gain and *PCDH17*) classified 38 out of 40 CO patients and 38 out of 40 NCO patients accurately (five-fold cross-validation mean 0.89 and 0.94 with 95% confidence interval 0.64 to 1, and 0.78 to 1, respectively),

rigorous risk prediction model building and validation that considers time, the spatial distribution of genomic features, and absolute risk quantification will require larger and independent cohort studies to translate these findings to clinical use.

**Fig. 6 Complex structural alterations in progression to ESAD. a** Plot shows count of biopsies with SV features in T1 and T2 biopsies in NCO and CO ($n = 80$ biopsies in each category of T1 NCO, T1 CO, T2 NCO, T2 CO). Comparison between T1 and T2 by Wilcoxon rank-sum test. **b** Proportion of SV events that are private to a single biopsy (grey) or shared between two or more biopsies (blue=NCO, orange=CO). (+) indicates significantly higher load of this SV feature is shared; (*) asterisk indicates a significantly higher load of SV feature is private, FDR 0.01 (Supplementary Data Fig. 9) **c** Total SV burden of NCO ($n = 40$), CO ($n = 40$) and ESAD ($n = 408$) patients, represented by the most rearranged sample per case. Statistical enrichment determined by Gamma-Poisson regression of SV burden as a function of group status, correcting for *TP53* mutational status. **d** Fraction of cases that harbor a complex amplification (BFB, double minute, and/or tyfonas), in NCO ($n = 40$), CO ($n = 40$) and ESAD ($n = 408$) (represented by presence of complex amplification type in any patient sample). Mann–Whitney U test was used for each comparison. **e** Left, UMAP clustering of NCO, CO, and ESAD patients using junction burden of most rearranged samples attributed to SV event types as input. Density contours determined on the basis of ESAD data only, with faded purple dots representing ESAD data points. Clusters determined by Gaussian mixture model regression. **f** Each sample harboring the SV feature of interest is represented by black dots in the scatterplot, while others are colored transparent grey. **g** Fraction of patients within each cluster containing complex amplification types. **h**, Fraction of NCO ($n = 40$), CO ($n = 40$), and ESAD ($n = 408$) cases within each cluster. Asterisks indicate the highest fraction is significantly higher than the other three (chi-square test, **** $P = 2.2 \times 10^{-16}$; *** $P = 0.0017$). Source data are indicated in Figure Source Data File.

## Discussion

We identified critical similarities and differences between NCO and CO patients that have implications for early cancer detection and risk stratification in BE patients. Genomes of purified BE epithelial cells, regardless of disease outcome, carry high levels of mutation, alterations in cancer-associated genes and pathways, and localized chromosome disruptions that are nonobligate to a cancer outcome. The inflammatory environment of the reflux-exposed esophagus likely influences the mutational processes that generate these somatic genomic changes[57,58], processes which are active before the clinical detection of BE and continue to evolve even in those who do not progress to ESAD. Multi-sample WGS revealed the critical distinction for individuals with BE who progress to ESAD is clonal expansion of *TP53* −/− cell populations and subsequent complex chromosomal structural events, which can be detected before the diagnosis of cancer and are highly specific to progression to ESAD.

The importance of *TP53* alterations to ESAD progression risk has been well established[9,18,33,44]. Our study design further refines our understanding, showing the clonal expansion of *TP53* −/− clones and development of complex structural variations are almost exclusive to CO patients and ESAD samples[23]. Focal clones with *TP53* mutations are common in the normal squamous esophagus[5] and localized *TP53* +/− clones can be detected in BE that does not progress to ESAD; thus, ultra-deep sequencing would likely identify minority cell populations with *TP53* mutations in many patients with BE, but miss the key cancer-promoting feature of clonal expansion of two-hit inactivated *TP53*. In this study, *TP53* −/− cell populations and complex SVs were detected in CO patients an average of 2.2 years prior to the diagnosis of ESAD (range 0.65 - 6.16 years). Only one NCO patient (ID 303) in our study had *TP53* −/− clones and subsequent complex SVs; however, this patient had high-grade dysplasia at the initial timepoint and died of other causes 2.81 years after T2, and likely represents a censored case that would have progressed to cancer given additional follow-up time. Any effective ESAD risk assessment in BE must be able to detect *TP53* −/− alterations or the complex SVs engendered by them.

A current goal of clinical management of BE is to incorporate somatic genomic-based cancer risk stratification to guide intervention strategies[11,24,59–61].The substantial spatial clonal heterogeneity observed in CO patients, with some biopsies indistinguishable from typical NCO biopsies, indicate a need for a multi-sample approach to early cancer detection or to ensure accurate ESAD risk assessment. Our finding of multi-clonal origins for the expanded BE segment in both CO and NCO based on levels of trunk mutations, as well as single-clonal origins, reflects this clonal heterogeneity occurring early during the establishment of the BE segment. Sub-clonal deconvolution and phylogenetic analysis of BE lineages have

shown the spatial clonal structure of the Barrett's segment remains relatively stable over time, even in patients who progress to cancer[62,63]. A recent low-pass WGS study in BE using FFPE samples[24] validated previous findings of ESAD risk associated with flow-cytometric DNA content abnormalities and SCA[11,19,49,53]. It identified a strong cancer risk signal from early time-point pooled biopsies, supporting an approach of a multi-sample, genomic-based ESAD risk assessment as an effective means of identifying patients most likely to benefit from targeted interventions.

Our results suggest nonprogressing BE resembles aging normal tissue, which undergoes stochastic alterations and selection in cancer-associated genes that confer survival advantages, and maintains the requisite cell- and tissue-based mechanisms that prevent the development of cancer. This hypothesis is supported by our data showing very similar mutation signatures, mutation loads and ESAD associated gene alterations in both NCO and CO, as well as by the results of mutagenesis analyses in mice[64]. *TP53* alterations arising early and carried along with the initial BE expansion may confer the highest ESAD risk[62,65]; alterations developing in isolation after the establishment of the segment would be constrained as those observed in other aging tissues and confer little cancer risk. We hope that other specific, objectively measured alterations, such as those involving methylation[66,67] or immune surveillance[66,68] might be combined with the genomic changes highlighted here to provide sensitive and specific ESAD risk prediction. This precancer dataset, derived from a natural history cohort with an ESAD endpoint, provides a valuable resource for examining questions about the evolution of ESAD at the genomic, cellular, and tissue levels.

## Methods

**Resources used in this study are listed in Supplementary Data File 31**
*Ethics approval and consent.* All research participants contributing clinical data and biospecimens to this study provided written informed consent, subject to oversight by the Fred Hutchinson Cancer Research Center IRB Committee D (Reg ID 5619).

*Study Design.* A case-control study was designed with 80 participants (Supplementary Data File 2) diagnosed with Barrett's esophagus (BE) selected from a larger case-cohort study[19] within the Seattle Barrett's Esophagus Program at the Fred Hutchinson Cancer Research Center. Inclusion criteria included ≥ two endoscopies each with ≥ two centimeter (cm) segment of BE (with a preference for ≥3 cm when possible) and none of the following interventions: endoscopic mucosal resection, radiofrequency ablation, photodynamic therapy, or previous esophageal surgical resection for cancer. Cases (cancer outcome, CO, $N = 40$) were defined as participants who progressed to diagnosis of esophageal adenocarcinoma (ESAD) during surveillance for early cancer detection. Controls (non-cancer outcome, NCO, $N = 40$) were defined as those who did not progress to ESAD during long-term follow-up, regardless of dysplasia grade. For each case, controls were randomly matched on baseline total somatic chromosomal alterations (SCA[19]), age at time point one (T1), time between T1 and time point two (T2), and gender (Supplementary Data Fig. 1). T1 was defined as either the baseline endoscopy in the Seattle Program ($N = 5$) or baseline+1 ($N = 75$), depending on tissue availability. T2 was

defined as the endoscopy at which ESAD was initially diagnosed for all but two CO, or the endoscopy selected for matched follow-up time for NCO. Biopsies from the diagnostic ESAD endoscopy were unavailable for CO patient ID 772 and lacked sufficient DNA for CO patient ID 568, so biopsies from the penultimate endoscopy were substituted. The demographics of this study are typical of BE and ESAD with 72 males and 8 females, with the average age at T1 of 65.5 years (range 43-82) for CO and 66.1 years (range 41-87) for NCO (Supplementary Data File 1).

*Sample processing.* Surveillance for early detection of cancer was performed using the Seattle Protocol[69]. In this study, for each time point two fresh-frozen endoscopic biopsies, independent of those used for histologic evaluation, were collected within the histologically defined regions of Barrett's esophageal tissue for WGS (Supplementary Data File 3). At each timepoint, one biopsy was selected in the BE segment at 1/3 and a second biopsy at 2/3 of the annotated distances from the gastroesophageal junction (GEJ). The biopsies used for WGS were not evaluated by histopathology. For CO, a biopsy from the same distance from the GEJ as the endoscopically detected ESAD was preferentially chosen as one of the two T2 biopsies. In 10 NCO, two biopsies from an additional time point (T3) with an average of 13.2 (range 7-17.7) years of follow-up from T1, were sequenced concurrently with that patient's T1 and T2 biopsies. From each biopsy, the Barrett's epithelial layer was isolated from the stroma as previously described[19]. DNA was extracted from the purified epithelium using Purelink Genomic DNA Kit (Invitrogen). Samples were processed and sequenced such that operators were blinded to patient outcome. Seven T1 or T2 biopsies and two T3 biopsies were in the lowest 5% of mutation load and lacked hallmarks of BE (Supplementary Data File 5). We hypothesize these biopsies may have little or no Barrett's epithelium, but in the absence of biopsy histology these have been included in all analyses unless otherwise indicated.

Normal controls from each participant were sequenced from either blood ($N = 62$) or normal gastric biopsies when blood was not available ($N = 18$) and analyzed by 30X WGS and 2.5 M SNP array for paired analysis. To test the reliability of the normal gastric sample as a control, an additional normal gastric biopsy was sequenced at 30X in seven of the 62 patients with a blood control available (Supplementary Data File 3). In these seven gastric samples only 27−43 (mean 34) somatic mutations per patient were detected, with only 0-2 mutations shared between the gastric sample and all BE biopsies, well within the range of noise, indicating gastric biopsies were reliable comparators at this depth of WGS analysis.

*2.5 M Illumina SNP array.* Somatic chromosome alterations (chromosome copy number and cnLOH) were assessed in BE and control biopsies using the Omni 2.5 M 8v1.3 array (Illumina) following manufacturer recommendations (Omni 2.5M-8 v1.2 for sample (391-23521-155R-31) at NYGC.

*WGS library preparation and direct sequencing.* For 78 patients (417 biopsies), WGS libraries were prepared using the TruseqDNA PCR-free Library Preparation Kit (Illumina) in accordance with the manufacturer's instructions at NYGC. Final libraries were evaluated using fluorescent-based assays including qPCR with the Universal KAPA Library Quantification Kit and Fragment Analyzer (Advanced Analytics) or BioAnalyzer (Agilent 2100). In two patients (ID 322 and ID 360), one or more biopsies had insufficient DNA to perform PCR-free library preparation and no replacements were available. Therefore, WGS libraries were prepared using the Illumina TruSeq Nano DNA Library Preparation Kit for all biopsies in these two patients ($N = 8$) and for their two normal control samples, in accordance with the manufacturer's instructions. Libraries were sequenced on an Illumina HiSeqX sequencer at $2 \times 150$ bp cycles using v2.5 chemistry. Sample QC preprocessing, DNA sequencing metrics and sample contamination, and normal concordance protocols are described in the Supplementary Methods.

*Sample purity and ploidy assessment.* Purity and ploidy for each sample were determined by a modified version of ASCAT (pASCAT v2.1)[70], with a gamma of 500. The accuracy of the calls of copy number change was determined by comparison to a custom algorithm[19]. In cases where the presence of multiple cell populations reduced the accuracy of the ASCAT calls, manual QC was performed that took into account the likelihood of aneuploidy cell populations (based upon the overall DNA content flow cytometric data from that patient (data not shown) as well as evaluation of allele-specific copy number differences and relatedness to other samples from the same patient with less ambiguity in their purity/ploidy determinations. In samples that displayed evidence of significant non-BE cell contamination (e.g., mixed with gastric cardia if the sample was near the GEJ), with a sharp variant allele frequency (VAF) distribution peak around 0.1 and low mutation load (2+ caller mutation load in lowest decile [<5,296 mutations]), purity values were derived from mutation-based algorithms (TITAN, NBIC-seq, and ASCAT).

*Genome Doubling.* Genome doubled samples were defined using the method of Bielski et al.[71], defined as samples with >50% of the genome with at least one allele with >=2 copies (cnLOH, gain, balanced gain, high gain) based on allele-specific copy number derived from pASCAT[70].

*SNV and INDEL calling and annotations.* Somatic SNVs were called by muTect v1.1.7[72], Strelka v1.0.14[73], and LoFreq v2.1.3a[74]. Indels were called using Strelka,

and somatic versions of Pindel v0.2.5[75] and Scalpel v0.5.3[76]. SNVs and indels were filtered using the default filtering criteria of each of the callers and only SNVs and indels that were called by at least two callers (2 +) were considered in this study (Supplementary Data File 38). Artifactual calls were removed by the use of a blacklist created by calling somatic variants on 16 random pairings of 80x/40x in-house sequenced HapMap WGS data. Mutations were annotated by SnpEff version 4.2 (build 2015-12-05) to assign each mutation to an Ensembl v75 transcript; "Functional" mutations were defined as those designated by SnpEff as high or moderate impact on protein function (see Supplementary Methods for details).

*Telomere length estimation.* To determine the average telomere length in each sample, a novel method called Telomeasure (https://github.com/nygenome/telomeasure) was developed that uses chromosome arm-level coverage to infer the number of telomeres and alignment to mock-human telomeres to estimate telomeric DNA content (see Supplementary Methods for details).

*Simple SV calling.* Simple SVs were called with Crest v1.0[77], Delly v0.6.1[78], and BreakDancer v1.4.0[79] (NYGC). Known germline SVs were removed and breakpoints used to generate a "high confidence" list of SVs. Only these high confidence SVs were considered in this study (Supplementary Data File 37).

Filtering and annotation of SVs was performed using bedtools (http://bedtools.readthedocs.org). SVs called by Crest, Delly, and BreakDancer were merged and annotated using BEDPE format. Two SV calls were merged if they shared at least 50% reciprocal overlap (for intrachromosomal SVs only), their predicted breakpoints were within 300 bp of each other and breakpoint strand orientation matched for both breakpoints. Thus, merging was done independent of which SV type was assigned by the SV caller. After merging, each SV was annotated with the closest CNV changepoint as detected by NBICseq[80] from sequencing read depth signals. This added confidence to true SV breakpoints that were not copy neutral. Additionally, an independent sensitive split read check was applied to each breakpoint using SplazerS[81]. SVs were retained if they were called by more than one tool or called by only one tool but also confirmed by 1) a CNV changepoint, or 2) at least three split reads in BE samples. All high-confidence Crest calls were retained due to the specificity of Crest-only high-confidence calls. All predicted copy number variants and SVs were annotated with gene overlap (RefSeq, Cancer Census) and potential effect on gene structure (e.g. disruptive, intronic, intergenic). See Supplementary Methods for additional details.

*Classes of SVs.* Using the methods described in Nik-Zainal et al.[55], SVs were binned into 32 SV classes: 5 size bins (0-10 kb, 10-100 kb, 100kb-1Mb, 1Mb-10Mb, >10 Mb) X (3 intrachromosomal SV types (*del, tds, inv*)) + 1 interchromosomal SV type (*trans*) X 2 cluster categories (yes/no). Details are described in Supplementary Methods.

*Assessment of Complex Chromosomal Events.* Complex chromosomal events (templated insertion chains (TIC), chromoplexy, chromothripsis, breakage fusion bridge cycles (BFB), double minutes, tyfonas, rigma, pyrgo) and simple deletions, duplications, inversions, translocations and inversion-duplications were also called by JaBbA[23]. Details are described in Supplementary Methods.

Briefly, algorithms and source code to identify the above events are available through R package, gGnome (https://github.com/mskilab/gGnome), in the functions "del" (deletions and rigma), "dup" (duplications and pyrgo), "simple" (translocations, inversions, and inverted duplications), "tic" (templated insertion chains), "chromoplexy", "chromothripsis", and "amp" (for BFB, double minutes, and tyfonas). Both junctions (non-reference edges in genome graphs) and genomic intervals (genome graph nodes) are marked as belonging to each event type. Further implementation details are found in[23].

To assess the pattern of sharing of junctions in complex SV events across samples, events for each type as determined using the above algorithms within a patient were overlapped across samples and collapsed into a union genomic footprint. Junctions corresponding to these events were determined as present or absent across all samples per patient. Sets of these junctions that were shared across a common set of samples were marked as a unique component of that event footprint. These components could then be placed on the phylogenetic trees as determined by somatic SNVs. Components present across all samples are placed at the root of the tree while those private to individual samples are placed at the tip. All other components, present in some, but not all, samples, are placed at tree branch points.

*Somatic chromosomal alterations (SCA).* SCA (copy number gains and losses, copy neutral LOH (cnLOH), homozygous deletion (HD), and high-level amplification) were measured by Illumina 2.5 M SNP array using allele-specific copy number with two algorithms (pASCAT[70] and Li et al.[19]) and by WGS with TITAN[82] and JaBbA[23]. HD were called only when both SCA calling algorithms agreed, and when the HD encompassed at least 50% of the gene coding sequence; in events with < 50% of the gene affected, each alteration was visually inspected using IGV[83] to verify HD loss of exonic sequences (Supplementary Data File 32). High-level amplification of a gene was defined as >5 chromosome copies spanning the gene in samples with no GD, and >6 copies in samples with GD, and only in instances

where both SNP-array and WGS calling methods called chromosome gain spanning the gene (see Supplementary Data File 12). See Supplementary Data File 37 for SCA segments and SCA by gene.

*Clustering of SV types by UMAP.* UMAP analysis was performed using the R package 'umap' available from: https://cran.r-project.org/web/packages/umap/. Default parameters were used with the exception of the following configurations which were set to: "n_epochs" = 1000; "metric" = "euclidean", "knn_repeats" = 5, and "random_state" = 10 for reproducibility.

## Statistical analysis of data

*General procedures.* Unless otherwise stated, all "mutation" analyses used only SNVs and indels called by at least two variant callers ("2+ caller"). All analyses used all T1 and T2 biopsies, including the anomalous biopsies; T3 biopsies were used only in a specifically indicated subset of experiments as T3 biopsies were available for 10 NCO patients only. Statistical analyses were carried out in SAS v9.4 unless otherwise indicated. Two-sided hypothesis tests were carried out for all the statistical analysis. When multiple comparisons were performed, p values were adjusted using the procedure of Benjamini-Hochberg, with the false discovery rates indicated in the main text. Mutations were considered "shared" if seen in more than one sample of a given patient and "private" if seen in only one sample; "subclonal" or "low VAF" indicates a VAF < 0.25 and "clonal" or "high VAF" a VAF > 0.25. In multivariable regression analysis, biopsies from the same patient were treated as repeated, not independent, samples from the same subject. A generalized linear model method (GLM)[84] was used to estimate the average population effect of groups if the response variable was continuous, such as mutation load; the generalized estimating equation (GEE) method was used for discrete response variables such as cancer outcome or *TP53* mutation status.

*Mutation load.* We assessed mutation load differences using GLM to test the difference between CO and NCO, controlling for patient age at sampling and distance from the GEJ. This test was performed on a per-biopsy basis, and per-patient basis (total number unique mutations per patient from four BE biopsies, see Supplementary Methods for details). VAF was contrasted between private and shared mutations in CO and NCO separately, for functional and nonfunctional mutations separately, using GLM analysis. The proportion of total mutations which were shared was contrasted between CO and NCO using GLM.

*Estimation of average change in mutation load between T1 and T2 for CO and NCO.* To assess the mutation load changes between T1 and T2 in patients with CO and NCO, we used EM (expectation and maximization) algorithms because 1) the mutation load within a given patient's esophagus at each timepoint is heterogeneous and 2) the T1 and T2 biopsies are not guaranteed to derive from the same clonal population. We evaluated the change of population distribution patterns of mutation load per biopsy in CO and NCO populations and at T1 and T2 separately. For each of the two populations, we used separate per biopsy mutation load data for T1 and T2 to run EM algorithms. Although mutation load per biopsy is highly heterogeneous, the overall data can be grouped into four groups; we therefore used four groups in the study. Let *nt1* and *nt2* be the number of groups identified in T1 and T2, *nt1* and *nt2* = 1, 2, 3, 4; and each with approximately normal distribution. Therefore, for T1 or T2 biopsy mutation load data of a given CO or NCO population, EM algorithm was used to identify $nt_i$ or $nt_j$ and their corresponding proportions (weights), with *i* and *j* being the *i*-th, *j*-th groups identified from T1 and T2 data of either patient in the CO or NCO groups. A bootstrap sampling method was used for each run. The proportion of each group for T1 and T2 data are $pt_{1i}$, and $pt_{2j}$ and mean mutation load are $mt_{1i}$, and $mt_{2j}$. The mutation difference between T1 and T2 are calculated by the formula $D = \sum_{i=1}^{nt1}\sum_{j=1}^{nt2} pt_{1i}pt_{2j}(mt_{1i}-mt_{2j})$. A total of 10,000 simulations with bootstrap sampling were run, and p values were calculated from the simulation results.

*Shannon Index for measuring diversity of somatic mutations.* Assume we sequenced $M_i$ molecules in locus *i*, and there are total *N* loci. Here we only count VAF, not NAF. Let $f_i$ be the VAF at locus *i*; the total number of variants is $f_i M_i$. The total molecules in the biopsy is $T = \sum_{i=1}^{N} f_i M_i$. Assuming all $M_i$ are equal to *M* in each locus, then $T = M * (\sum_{i=1}^{N} f_i)$. Therefore the Shannon diversity is calculated as $H = -\sum_{i=1}^{N}(f_i M_i)/(M(\sum_{i=1}^{N} f_i))\log((f_i M_i)/(M(\sum_{i=1}^{N} f_i)))$. Since we assume $M_i = M$, then $H = -\sum_{i=1}^{N} f_i/(\sum_{i=1}^{N} f_i)\log(f_i/(\sum_{i=1}^{N} f_i))$.

*Divergence between biopsies.* For each pair of biopsies, the number of mutations not shared by the paired biopsies was counted for divergence calculation[85]. We measured pairwise divergence between biopsies and tested for association with progression status using GLM for all mutations, and separately for functional, nonfunctional, clonal, and subclonal mutations.

*Mutation Signatures.* The complete set of single nucleotide variants (SNVs) called by at least two somatic mutation calling algorithms across all samples was classified in a matrix with 96 mutational channels using SigProfilerMatrixGenerator[86].

Analysis of de novo mutational signatures was performed by applying SigProfiler[29] for 1 to 30 signatures with each extraction being performed for 500 iterations. A total of eight de novo signatures were identified in the examined samples (see Supplementary Methods for details). These eight de novo signatures were best matched to a set of 10 previously reported COSMIC signatures[86]. 6 of 8 de novo signatures had an exact match with the COSMIC signatures SBS1, SBS3, SBS5, SBS17a, SBS17b, and SBS40; another was a mixture of SBS2 and SBS13; and one was a mixture of SBS18 and low levels of SBS34. The activity of each COSMIC signature in each sample was quantified as previously performed[86]. Hierarchical clustering based on cosine similarity was performed using all signatures across all 320 T1 and T2 samples and 80 patients.

*Phylogenetic Analysis.* We constructed maximum parsimony trees of the four biopsies per patient based on SNVs only, using *dnapars* from Phylip v3.695[87]. Support for the trees was determined based on 1000 bootstrap replicates made using *seqboot* and *consense* from Phylip v3.695[88]. For details see Supplementary Methods.

*SV Categories.* We categorized SV events into 32 classes based on type, size, and event clustering[55]. Clustered events were infrequent in our dataset and therefore were collapsed into a single class as the sum of clustered events. Together with the 16 non-clustered events, a total 17 classes of SV categories were compared between CO and NCO using Kruskal-Wallis test.

*Frequencies of mutations in individual loci and mutation-selection analysis.* The frequencies of the functional mutations in each gene in 40 CO and 40 NCO were compared using two-sided Fisher's exact test. The individual testing p values were filtered as described in Tuglus and van der Laan[89] and corrected for multiple testing using the Benjamini-Hochberg method[90].

We inferred the action of positive selection using dN/dScv v0.0.1.0[37] separately in NCO and CO on total mutations. We tested whether mutations in loci identified as under positive selection were preferentially shared or private using GLM.

*Pathway analysis.* Pathways from multiple sources were curated as shown in Supplementary Data File 16, yielding 6,293 loci in 312 pathways; note that any given locus could appear in multiple pathways. For this analysis we considered a gene to be altered if it contained functional SNV or indel mutations or if at least 50% of the coding sequence was homozygously deleted. Given the high level of uncertainty of the effects of copy number change and SVs on gene expression and function, these alterations were not included in this analysis. We tested for association between CO or NCO status and frequencies of altered genes in each pathway by Fisher's exact test for both the number of pathway mutations per patient and the number of patients with one or more pathway mutations. Benjamini–Hochberg was used for multiple comparisons *p* value correction. We also repeated these analyses disregarding mutations in *TP53*, due to its strong individual association with progression and its presence in a large number of different pathways.

*TP53 analyses.* For classifying *TP53* status, a "mutation" was defined as any high or moderate impact SNV or indel (called by SNPeff), HD affecting at least one exon of *TP53*, or SV affecting the *TP53* coding sequence or splice sites. SCA affecting *TP53* had to span at least 50% of *TP53* exonic regions. *TP53* status by the patient was based on the sample with the maximum *TP53* call. If more than one alteration was present in a sample, the sample was categorized based on the alteration that resulted in the most severe outcome. If more than one sample per patient had alterations, the patient was categorized based on the alteration that resulted in the most severe outcome. All *TP53* alterations were manually verified using IGV and SCA was verified using Partek. Each sample was classified as zero-hit (+/+, no *TP53* alteration), one-hit (+/−, evidence for *TP53* alteration affecting a single allele or alteration in a minority clone), or two-hits (−/−, evidence for alteration affecting both *TP53* alleles). Pathogenicity of *TP53* mutations was assessed based upon IARC *TP53* mutation database[48], and by identifying the domain affected by each mutation. Differences in VAF among *TP53* mutations in NCO and CO were evaluated per biopsy using GLM, and differences in the number of biopsies containing a *TP53* mutation per patient were evaluated using Fisher's exact test. We used a nonparametric method (Kruskal-Wallis test) to test SCA and SV loads in *TP53* mutant biopsies vs.*TP53* wild-type in each mutation load strata. GLM was used to test for an increase of point mutation load and for a trend toward increasing Mb of SCA and count of SVs. GLM was used to test for association between progression status or TP53 mutation status (dependent variable) and telomere length (independent variable).

*Differently mutated genes with SCA and SV load.* Functional SNV and Indel gene mutations were quantified according to categories of SCA, SV load and GD in CO and NCO patients. We used the 95% confidence intervals of SCA load (103.9 Mb) and SV counts (71) in NCO samples without any *TP53* alterations as the threshold to divide biopsies into three groups: Group1 - low SCA/SV (SCA <=103.9 Mb and SV <=71); Group 2 - moderate SCA/SV (SCA between 103.9 Mb and 1500 Mb or SV > 71 and SCA < 1500 Mb); and Group 3—genome doubled with SCA > 1500 Mb

(regardless of SV counts). We contrasted the frequencies of functional SNVs for individual genes between the three groups using GLM.

*Significant somatic genomic alterations and their relationship with EA risk.* Given the large number of alterations, particularly those that occur significantly more often in CO patients but still at a low absolute frequency, overfitting of prediction models is an issue that must be considered. Therefore we first identified 47 genomic alterations shown to occur significantly more frequently in CO patients as candidates for EA risk prediction markers (Supplementary Data File 30). We first used LASSO (least absolute shrinkage and selection operator[91]) to select a smaller number of markers from the 47 markers with five-fold cross validation. The leave-20%-out sampling LASSO procedure identified 14 markers that were frequently selected. We then used traditional forward selection regression methods to evaluate the relationship of the most frequently selected markers. Since *TP53* alteration is the most predictive and robust marker for EA risk prediction, we first included *TP53* one- and two-hits in the simple regression model and then added more markers from the 14 markers mentioned above. This approach generated the six marker model, all of which have significant coefficients, described in the results.

**Reporting summary**. Further information on research design is available in the Nature Research Reporting Summary linked to this article.

## Data availability

The molecular data generated in this study - binary alignment matrix (BAM) files of whole genome sequence reads on a reference genome and Infinium 2.5 SNP array idat files – have been deposited at the NCBI dbGaP database under accession code phs001912.v1.p1 [https://www.ncbi.nlm.nih.gov/projects/gap/cgi-bin/study.cgi?study_id=phs001912.v1.p1]. The dbGap data are available under restricted access for research in Barrett's esophagus or esophageal adenocarcinoma; access can be obtained by requesting Authorized Access (Individual Level Data and SRA Data) through the dbGaP Authorized Access System upon approval of the Data Access Request (DAR): https://dbgap.ncbi.nlm.nih.gov/aa/wga.cgi?login=&page=login. Individual-level data are available for download by authorized investigators: https://view.ncbi.nlm.nih.gov/dbgap-controlled. Data dictionaries and variable summaries are available on the dbGaP FTP site: https://ftp.ncbi.nlm.nih.gov/dbgap/s. The public summary-level phenotype data may be browsed at the dbGaP study report page: https://www.ncbi.nlm.nih.gov/projects/gap/cgi-bin/study.cgi?study_id=phs001912.v1.p1. Individual patient data are protected and are not available due to data privacy laws. The remaining processed data are available within the Article, Supplementary Information and Supplementary Data files. Source data are provided with this paper.

## Code availability

Code for the program Telomeasure can be found at https://github.com/nygenome/telomeasure.

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

## Acknowledgements

This study was made possible by the lifelong commitment of Dr. Brian Reid to improving the lives of patients with Barrett's esophagus. We thank all the research participants who have made this study possible. We thank and acknowledge the support of the following medical and research staff who provided clinical care of the Seattle Barrett's Esophagus Program patients over the course of three decades and thus made this study possible: Dr. Douglas Levine, Dr. Peter Rabinovitch, Dr. Thomas Vaughan, Dr. Patricia Blount, Dr. Kamran Ayub, Dr. Shayan Irani, Dr. Michael Saunders, Dr. Rodger Haggitt, Dr. Robert Odze, Susan Irvine RN, Christine Karlsen, and Patricia Christopherson. David Cowan and Terri Watson provided database support. Tami Tolpa (Fig. 1), Pritha Chanana, Timothy Chu, and Michael Northup helped with figure preparation. Jon Yamato and Diego Mallo provided additional analytical support. Amber Karnofski provided administrative support. T.G.P., P.C.G., K.M.O., C.A.S., B.J.R., and X.L. were supported by NIH P01 CA91955 and P30 CA015704. M.K.K. and L.P.S. were supported by NIH P01 CA91955. C.C.M. was supported by NIH grants P01 CA91955, U54 CA217376, U2C CA233254, R01 CA140657 and CDMRP Breast Cancer Research Program Award BC132057. Marcin Imielinski was supported by Burroughs Wellcome Fund Career Award for Medical Scientists, Doris Duke Clinical Foundation Clinical Scientist Career Development Award, Starr Career Consortium Award, Melanoma Research Alliance Team Science Award, and NIH U24-CA15020. Kevin Hadi was supported by NIH/NCI F31 CA232465 Graduate Research Fellowship.

## Author contributions

T.G.P., P.C.G., C.A.S., B.J.R., and X.L. conceived of and designed the study. P.C.G., C.A.S., and B.J.R. provided patient samples and flow-cytometric, histology, and epidemiology annotations. TGP and CAS performed epithelial isolation and DNA extraction. L.P., K.H., M.S., K.A., J.S., M.J., A.C., X.Y., R.S., A-K.M., and X.L. performed sequencing and or SNP-array experiments, and/or implemented variant calling software. TGP, PCG, CAS, MKK, LPS, KH, XY, MI, XL developed and designed methodology and created models. K.M.O., M.K.K., L.P.S., K.H., M.S., K.A., J.S., X.Y., A-K.M., M.I., and X.L. developed and implemented software. T.G.P., P.C.G., C.A.S., M.K.K., L.P.S., K.H., K.A., M.S., X.Y., and X.L. designed and conducted validation experiments. K.M.O., M.K.K., L.P.S., K.H., and X.L. performed formal analytics and statistical analysis. K.M.O., L.P.S., M.K.K., K.H., M.S., K.A., R.S., J.S., M.J., A.C., M.J., A.C., X.Y., M.I., N.R., and X.L. provided instrumentation, computer resources, and analysis tools. T.G.P., P.C.G., K.M.O., C.A.S., K.H., R.S., M.S., K.A., J.S., and X.Y. performed data curation. T.G.P., P.C.G., K.M.O., C.A.S., M.K.K., L.P.S., B.J.R., and X.L. prepared the initial draft of the manuscript. T.G.P., P.C.G., C.A.S., M.K.K., C.C.M., M.I., N.R., B.J.R., and X.L. provided critical review and revisions of the manuscript. T.G.P., P.C.G., K.M.O., L.P.S., K.H., M.S., K.A., A.C., X.Y., R.S., and X.L. conceived of and created data visualization for figures. T.G.P., P.C.G., C.A.S., M.I., N.R., B.J.R., and X.L. provided supervision, oversight and leadership for the management, coordination, planning and execution of this study. P.C.G., E.V., and B.H. provided management for research activity planning and execution. T.G.P., P.C.G., C.A.S., M.I., B.J.R., and X.L. acquired financial support for the project leading to this publication. All authors approved the final version of the manuscript.

## Competing interests

The authors declare no competing interests.
