## [Peer review file · Nature Communications]

REVIEWER COMMENTS

Reviewer #1 (Remarks to the Author): Expert in BE and ESAD genomics, cancer evolution, and computational genomics

The authors present a novel, comprehensive, well designed and thoroughly conducted analysis of the genomic events distinguishing Barrett's Esophagus tissue from individuals with non-progressive lesions (NCO) versus ones who have cancer outcomes (CO). This is an important study for the field and highlights some key genomic biomarkers that could be used to predict the risk of esophageal cancer development. The authors show that differences between NCO and CO lie not at the level of SNVs/indels and mutational processes, but at the level of p53 double hits, larger structural variation (including rigma and BFB) and genome doubling. Their multiregion sequencing approach also highlights the existence of biopsies that are indistinguishable genomically between NCO and CO, which brings forward the important message that multiregional sequencing may be needed to assess risk of progression to esophageal cancer. The statistical analyses performed were appropriate.

I have the following suggestions and questions:

1. The authors should show the statistics supporting their choice for the number of signatures in the mutational signature analysis.
2. The pathway analysis was only performed considering SNVs and indels. What would the differences observed be if copy number and structural alterations were also taken into account?
3. Cluster 4 in Figure 6e is remarkably distinct. Is the apparent increase in double minute events sufficient to explain this? How robust are the clusters uncovered? Were all the cancer samples therapy naïve, and if not, could therapy-induced alterations impact the clustering?
4. Were there any driver gene events associated with any of the SV clusters from Fig 6e?
5. Was there any correlation between SV clusters uncovered and clinical outcome (for the cancer patients only)?
6. Differences between T1 and T2 samples are listed throughout the manuscript, but it would be helpful to have a summary figure with the number and type of variants (particularly copy number and SV, which are less clear than SNVs) shared and unique between NCO/T1 and NCO/T2, and CO/T1 and CO/T2 across individuals.
7. For the regression-based modelling of NCO/CO outcomes, the authors evaluated if one-hit TP53, interacting with the risk features PCHD17 and chr18:18-25Mb gains, could add ESAD risk prediction power independently of the two-hit p53. Have the authors considered testing other features besides these two that might distinguish NCO/CO in the case of single hit TP53 mutation?

8. How would the simple regression model employed compare to a lasso approach?
9. Minor point: the Shannon diversity index explanation in the methods is included as an image rather than text, which is unusual.

Reviewer #2 (Remarks to the Author): Expert in BE genetics and gastroenterology

This study presents results of whole genome sequencing of BE biopsies in a unique prospective cohort confirming other recent studies including one published by Rebecca Fitzgerald's group in Nature Genetics showing that CNV/aneuploidy occur early before the development of Cancer and may be the best known molecular marker to predict progression. Previous studies have also noted a high rate of somatic mutations in BE as found in this study. This study also confirms the fact that cancers do not necessarily develop from the same clones as found in the BE. Adam Bass's studies have shown that the cancers may develop from different clones and shown similar branching of the tree of multiple clones as found in this study. It adds to the growing literature on the complexity of carcinogenesis and provides a more detailed picture of the clonal diversity and expansion theory that the authors have championed.

1. Biopsy sampling of BE is well known to be problematic as the investigators have shown in previous studies. How were the biopsies selected for sequencing in terms of the location of the biopsy? Was there an ability to correlate the clonality of the biopsy with the subsequent cancer? Could they learn more by looking at other biopsies on these patients that varied temporally?
2. Can they expand and explain their definition of spatially mapped biopsy? If I understand correctly they had four biopsies per level and multiple levels depending on length of the BE segment. Am I interpreting their study correctly? Or did they just take four biopsies from one level? Was it possible to associate the location of the BE biopsy with the location of the cancer?
3. The large number of mutations and clones could simply represent a tissue that is mutating in a highly inflammatory milieu. Other than p53 mutations are there any other clues regarding drivers? The GATA6 findings are intriguing and also supported by other studies.
4. The Single Base Signature analysis does not add much to the study. Neither does the evaluation of fragility. These sections could be shortened.
5. The examination of somatic mutations in genes also repeats previous studies.

We would like to thank the editors and reviewers for their encouraging and helpful comments about our manuscript and for giving us the opportunity to address their concerns in a resubmitted version. The reviewer's comments and suggestions have made this a better manuscript. We address the individual concerns of the reviewers point by point below, with the reviewers comments in bold, followed by our responses and outlining any specific changes proposed in the manuscript. All changes in the manuscript are highlighted in yellow in the main manuscript file.

Reviewer #1 (Remarks to the Author): Expert in BE and ESAD genomics, cancer evolution, and computational genomics

The authors present a novel, comprehensive, well designed and thoroughly conducted analysis of the genomic events distinguishing Barrett's Esophagus tissue from individuals with non-progressive lesions (NCO) versus ones who have cancer outcomes (CO). This is an important study for the field and highlights some key genomic biomarkers that could be used to predict the risk of esophageal cancer development. The authors show that differences between NCO and CO lie not at the level of SNVs/indels and mutational processes, but at the level of p53 double hits, larger structural variation (including rigma and BFB) and genome doubling. Their multiregion sequencing approach also highlights the existence of biopsies that are indistinguishable genomically between NCO and CO, which brings forward the important message that multiregional sequencing may be needed to assess risk of progression to esophageal cancer. The statistical analyses performed were appropriate.

We thank the reviewer for their kind comments.

I have the following suggestions and questions:

1. The authors should show the statistics supporting their choice for the number of signatures in the mutational signature analysis. SigProfiler was utilized to identify the number of operative mutational signatures across the examined BE samples. The tool identified solutions with 1, 2, 3, 4, 5, 6, 8 and 9 operative signatures as stable and reproducible (minimum stability >0.80; Alexandrov, et al., 2020). Statistically, the solution with 9 signatures did not provide a better description of the data compared to the solution with 8 signatures ($P = 0.0976$) but it did describe the data better than using 7 signatures ($P = 3.77E-19$). SigProfiler selected 8 signatures as the optimal solution since this is a stable solution which describes the data just as well as the stable solution with maximum number of signatures. We have included a supplementary table (Supplementary Table 35, Mutation Signature Analysis Solutions) showing the statistics for selecting the number of mutational signatures based on the somatic mutations identified in these BE samples and have added the following in the extended methods:

“Mutation Signature Analysis Solutions

SigProfiler was utilized to identify the number of operative mutational signatures across the examined BE samples (Supplementary Table 35). The tool identified solutions with 1, 2, 3, 4, 5,

6, 8 and 9 operative signatures as stable and reproducible (minimum stability >0.80(Alexandrov et al. 2020)). Statistically, the solution with 9 signatures did not provide a better description of the data compared to the solution with 8 signatures (p-value: 0.0976) but it did describe the data better than using 7 signatures (p-value: 3.77E-19). SigProfiler selected 8 signatures as the optimal solution since this is a stable solution which describes the data just as well as the stable solution with maximum number of signatures.”

2. The pathway analysis was only performed considering SNVs and indels. What would the differences observed be if copy number and structural alterations were also taken into account? We agree with the reviewer that determining disrupted pathways using all available types of genomic alterations would potentially be a better approach. The difficulty of including copy number and structural alterations (other than the double deletions that we did include), is their lack of gene specificity and ambiguous phenotypic effect. SNVs, indels and double deletions were chosen as being the least ambiguous with respect to their effects on gene function. We employed a cautious approach, requiring 2+ callers for each SNV and indel and requiring a designation of “high” or “moderate” impact on protein function as determined by SnpEff to identify alterations likely to affect gene function with high confidence. As well, we required >50% of the coding sequence to be lost by homozygous deletion to have high confidence that normal gene function would be disrupted. In contrast, many, if not most, copy number changes affect multiple genes. Furthermore, determining the phenotypic effects of copy number alterations and SVs is problematic; for example, the effect of a change in copy number from 2 to 3 versus a change from 4 to 3 (e.g., in a patient that has undergone genome doubling) could be very different even though the end copy number (3) is the same. What objective metrics can be used to determine the effect of SVs, which can disrupt coding sequence and/or amplify to different degrees, on gene function? As well, it is not currently possible to determine the effect of a copy number change in a single gene that occurs as part of a large amplification events or a genome doubling. Given the high level of uncertainty of the effects of copy number change and SVs on gene expression and function, we thought it prudent to leave them out of the analysis, with the hope that better analytical pipelines in the future may be able to more definitively address this question. We have added this reasoning in the pathway section and in the methods describing the pathway analysis: “We conducted a comprehensive gene pathway analysis to evaluate the effects of the individual mutations (SNVs/indels and double deletions) on pathway function (Supplementary Tables 16, 17, and 18); copy number and SV alterations were not included as their effects on gene function are more difficult to determine.”

3. Cluster 4 in Figure 6e is remarkably distinct. Is the apparent increase in double minute events sufficient to explain this? How robust are the clusters uncovered? Were all the cancer samples therapy naïve, and if not, could therapy-induced alterations impact the clustering?

We examined cluster robustness in a variety of ways. Cluster likelihood analysis indicated an elbow at 4 clusters (meaning the majority of the information from this data set was represented by 4 clusters), whereas cluster stability analysis using bootstrapping suggested 5 clusters had higher overall stability. However, two of these five clusters had lower Jaccard stability indices, with further analysis indicating these two clusters represented a split of the existing Cluster 3

into two smaller, less stable clusters. Therefore, we used 4 clusters as the most stable representation of the data.

The separation of Cluster 4 away from the rest of the samples appears to be driven primarily by a higher frequency of double minutes and a much higher frequency of simple translocations (something also observed in Cluster 2 (see Fig. 6f)). While the small number of BE patients in Cluster 4 make it difficult to identify the biological reasons for the clustering (e.g., perhaps these patients have cell populations with their genomes in an open chromatin configuration and are more likely to undergo translocations (Hogenbirk, et al., PNAS 2016)), these data show the genomic changes seen in advanced ESAD samples are detectable in patients well prior to a diagnosis of cancer.

The question of therapy induced alterations is an important one. Of the 408 ESAD cases that were analyzed for Figure 6, 401 were treatment naive; of the 7 cases that underwent chemotherapy, only one was found in cluster 4 (with two in cluster 1, and 4 in cluster 3), suggesting the clustering was not likely to have significantly affected by the small number of cases that had undergone therapy. We have modified the manuscript in the section “Development of complex chromosomal structural variants during progression to ESAD”, to state “Using the same SV calling pipeline, we compared the SV features between NCO and CO with data from 408 ESAD cases^{13,23}, the vast majority of which (401/408) were treatment naive.”

4. Were there any driver gene events associated with any of the SV clusters from Fig 6e?

We examined if alterations in any of the 127 ESAD associated genes were significantly associated with a particular cluster. Mutations in *TP53* and amplification of *CCNE1* (cyclin E1) were found to occur at a significantly higher frequency in Cluster 3 than in the other clusters. We have added the following sentence to this section - “Mutations in *TP53* and amplification of *CCNE1* (cyclin E1) were the only ESAD gene alterations found at a significantly higher frequency in a given cluster (both higher in Cluster 3, $P=0.0014$ and $P=0.001$, respectively).”

5. Was there any correlation between SV clusters uncovered and clinical outcome (for the cancer patients only)?

No significant difference in overall survival was found for ESAD patients in different clusters. We added this information to the manuscript (“No significant difference in overall survival was found for ESAD patients in different clusters.”)

6. Differences between T1 and T2 samples are listed throughout the manuscript, but it would be helpful to have a summary figure with the number and type of variants (particularly copy number and SV, which are less clear than SNVs) shared and unique between NCO/T1 and NCO/T2, and CO/T1 and CO/T2 across individuals.

This is an interesting question, but a difficult one to answer given the number of variable characteristics of SV (e.g., breakpoint locations) and SCA events (e.g., breakpoints and number of copies). We have created Extended Data Figure 10, which shows comparisons between the most altered T1 and T2 samples for a number of SNV, SV and copy number characteristics (Part A). Given the complexity and heterogeneity of copy number changes, including a need to determine which allele is being altered, it is more difficult to give definitive measures of shared and unique copy number change events between the two time points. Extended Data Figure 10 also shows how frequently SVs were shared across time points (Part B); in each graph SV types and the number of times that SV was observed in either NCO or CO patients is shown in the columns, while the proportion of events that were private to a single biopsy or shared either across time points or only across space within a time point are shown. We have added a reference to this figure in the text ; “In both NCO and CO, γ was the only complex SV feature detected significantly more often as a shared (early) event ($P < 0.0001$)(Fig. 6b, Supplementary Table 28, Extended Data Fig. 9); while maximum measures of simple and complex SVs were generally higher at T2 (Extended Data Fig. 10), all other SV features are more likely to be either private events or have no specific temporal pattern.”

7. For the regression-based modelling of NCO/CO outcomes, the authors evaluated if one-hit TP53, interacting with the risk features PCHD17 and chr18:18-25Mb gains, could add ESAD risk prediction power independently of the two-hit p53. Have the authors considered testing other features besides these two that might distinguish NCO/CO in the case of single hit TP53 mutation?

We did indeed evaluate if any of the 14 risk features identified by LASSO interacted with one-hit *TP53* to provide ESAD risk prediction power independent of two-hit *TP53* (see also the response to #8 below). Only chromosome 18:18-25mb gains interacting with one-hit *TP53* had statistically significant predictive power independent of two-hit *TP53*. We have made this more clear in the first paragraph of this section.

8. How would the simple regression model employed compare to a lasso approach?

We thank the reviewer for pointing out the need for more clarity in our evaluation of features that distinguish NCO from CO. Given the large number of potential features, and the relatively modest size of our study (40 CO and 40 NCO), we did not feel that simply using a LASSO approach would robustly identify features that would discriminate CO from NCO; therefore we used LASSO as a preliminary step to reduce the number of potential features and then performed a regression analysis. Univariate analysis of SNV/indel, SV and copy number alteration data initially identified 47 features significantly associated with CO (Supplementary Table 30). We then used LASSO to reduce the number of potential features out of this group. Using a random sampling training and cross validation procedure that left out 20% of samples for testing in each iteration, we identified the 14 features noted in the manuscript that were most frequently selected in the prediction models. After LASSO, we carried out the regression analysis based on these 14 features to evaluate the relationships among them in detail, including investigation on the effects of *TP53* gene status (1 or 2 hit) with consideration of VAF on EA risk outcome. With *TP53* (1 or 2 hit) considered in the regression model, we then added

other significant genes (identified by univariate analysis) one by one to test if EA risk could be better explained by the simple model. To clarify how this analysis was performed, we have modified the first paragraph of this section of the manuscript to clarify our approach and added a detailed paragraph in the methods describing this procedure.

“Identifying markers of risk for progression of BE to ESAD is a key component for improving patient care. Given the sample size of our study, the complexity of the somatic genome, and with a vast number of features occurring at low frequencies, we used a multi-step approach to identify robust markers associated with CO patients. First, univariate analyses of individual SNV/indel, SV and copy number alteration data identified 47 features significantly associated with CO, including *TP53* alterations (one- and two-hit), GD, chromosome 18:18-25Mb gain, complex SV features (BFB, DM, tyfonas), ESAD associated genes with significantly higher frequency alterations in CO, genes with significantly higher functional mutations in CO, mutated genes associated with SV features and GD, and ESAD genes with high-level amplification (Supplementary Table 30). Next, we used a LASSO approach to reduce the number of potential features out of this group. Using a random sampling training and cross validation procedure that left out 20% of samples for testing in each iteration, we identified the 14 features that were most frequently selected in the prediction models (Supplementary Table 30).”

9. Minor point: the Shannon diversity index explanation in the methods is included as an image rather than text, which is unusual. We have corrected the mathematical formulas in the manuscript.

Reviewer #2 (Remarks to the Author): Expert in BE genetics and gastroenterology

This study presents results of whole genome sequencing of BE biopsies in a unique prospective cohort confirming other recent studies including one published by Rebecca Fitzgerald’s group in Nature Genetics showing that CNV/aneuploidy occur early before the development of Cancer and may be the best known molecular marker to predict progression. Previous studies have also noted a high rate of somatic mutations in BE as found in this study. This study also confirms the fact that cancers do not necessarily develop from the same clones as found in the BE. Adam Bass’s studies have shown that the cancers may develop from different clones and shown similar branching of the tree of multiple clones as found in this study. It adds to the growing literature on the complexity of carcinogenesis and provides a more detailed picture of the clonal diversity and expansion theory that the authors have championed.

We thank the reviewer for their kind comments

1. Biopsy sampling of BE is well known to be problematic as the investigators have shown in previous studies. How were the biopsies selected for sequencing in terms of the location of the biopsy?

We chose to sample at $\frac{1}{3}$ and $\frac{2}{3}$ of the length of the Barrett's segment so as to make sure the samples were well separated in space, but not so close to the border of the segment that they might land in non-Barrett's tissue. Samples were not evaluated by histopathology and therefore were histology independent. We have made this more clear in the modified methods paragraph that is included in the response to point 2 below.

Was there an ability to correlate the clonality of the biopsy with the subsequent cancer?

Unfortunately, the cancers themselves were not analyzed for this study since most of them were detected when still microscopic. In order to maximize the chance that one of the samples would be near the location of the diagnosed cancer, one of the samples at T2 in the CO patients was taken from the same level at which the cancer was diagnosed; however, given the heterogeneity of the BE segment, we can't be sure to be sampling the same clonal population that developed into the cancer.

Could they learn more by looking at other biopsies on these patients that varied temporally?

Undoubtedly, more could be learned from more intensive temporal sampling. Future studies will be required to more accurately estimate how far in advance we can detect high risk clones, predictive of progression to cancer. More extensive temporal sampling would also allow us to detect changes in the mutation rate in cell lineages. As sequencing costs continue to decrease, we hope to be able to perform such extended studies in the future.

2. Can they expand and explain their definition of spatially mapped biopsy? If I understand correctly they had four biopsies per level and multiple levels depending on length of the BE segment. Am I interpreting their study correctly? Or did they just take four biopsies from one level? Was it possible to associate the location of the BE biopsy with the location of the cancer?

We apologize if we weren't clear enough concerning the selection of the biopsies for this study. Endoscopic surveillance for early detection of cancer was based on taking four biopsies per level, every 1-2 cm depending on risk status of the patient using dysplasia grading. Those clinical biopsies were formalin fixed and evaluated by histopathology. However, the biopsies examined in this WGS study are independent of those taken for histopathology; the two biopsies taken at different levels at each of two different timepoints for WGS were a separate set of research biopsies that were frozen rather than fixed. They are spatially mapped in that we know the level in the esophagus at which they were taken, although the circumferential position in the esophagus was not known.

We agree with the reviewer that the heterogeneity of the BE segment can lead to sampling issues and this is one of the points of our study. Given the reviewer's questions in both points 1 and 2 above, we have modified the section in the methods describing the selection of biopsies for sequencing. The new version reads:

"Surveillance for early detection of cancer was performed using the Seattle Protocol⁶⁸. In this study, for each time point two fresh-frozen endoscopic biopsies, independent of those used for

histologic evaluation, were collected within the histologically defined regions of Barrett's esophageal tissue for WGS (Supplementary Table 3). At each timepoint, one biopsy was selected in the BE segment at $\frac{1}{3}$ and a second biopsy at $\frac{2}{3}$ of the annotated distances from the gastroesophageal junction (GEJ). The biopsies used for WGS were not evaluated by histopathology. For CO, a biopsy from the same distance from the GEJ as the endoscopically detected ESAD was preferentially chosen as one of the two T2 biopsies. In 10 NCO, two biopsies from an additional time point (T3) with an average of 13.2 (range 7-17.7) years of follow-up from T1, were sequenced concurrently with that patient's T1 and T2 biopsies."

3. The large number of mutations and clones could simply represent a tissue that is mutating in a highly inflammatory milieu. Other than p53 mutations are there any other clues regarding drivers? The GATA6 findings are intriguing and also supported by other studies. We agree with the reviewer that the highly inflammatory environment of the reflux exposed esophagus likely contributes to the high levels of mutation found in BE samples. The higher frequencies of mutational signature SBS1, which can reflect the increased cell turnover required for tissue repair and mutational signature SBS18, associated with exposure to reactive oxygen species, support this idea that inflammation and associated cellular damage occur in BE tissues. However, the fact that these signatures do not predominate and that other signatures, such as SBS17a and 17b which have unknown etiology but are frequently observed in BE and ESAD at high levels, suggest a more complex explanation for the large number of mutations observed. We have modified the manuscript in the first paragraph of the discussion to emphasize the potential role of the inflammatory environment with the following: "The inflammatory environment of the reflux-exposed esophagus likely influence the mutational processes that generate these somatic genomic changes^{58,59}, processes which are active before the clinical detection of BE and continue to evolve even in those who do not progress to ESAD."

Our results on alterations in driver genes suggest many of the genes categorized as driver genes are also found altered at similar frequencies in NCO patients, suggesting they are selected during the development of BE instead of being obligate for development of cancer. Other genes are altered in a small percentage of patients, consistent with similar analyses reported in ESAD patients. *TP53* is clearly an exception, with data from multiple studies supporting its key role in carcinogenesis, and we were able to expand on this by our analysis of multi-hit/clonally expanded *TP53* alterations and their association with progression.

4. The Single Base Signature analysis does not add much to the study. Neither does the evaluation of fragility. These sections could be shortened.

The single base signature analysis showed, contrary to initial hypotheses, that there is no distinct mutational driver of progression to cancer in BE patients that might be targeted to prevent progression, an important point to make to the researchers in the field. It is also important for understanding the types of mutational processes that occur in both progressing and nonprogressing BE, which addresses the reviewer's question 3 above. While we think these are important insights, we have shortened the section as suggested.

5. The examination of somatic mutations in genes also repeats previous studies. While the mutation signature and somatic mutation analyses are not novel, the characterization of these measures of genomic change in matched populations of patients who did or did not progress to cancer in multiple samples over both time and space is a unique approach that provides important data on premalignancy. We have identified which findings confirm previous studies' results, and which are novel to our study.

REVIEWERS' COMMENTS

Reviewer #1 (Remarks to the Author):

The authors have addressed all my comments in a suitable manner. I recommend the manuscript for publication.

Reviewer #2 (Remarks to the Author):

Have answered my comments